# Unraveling the Key Components of OOD Generalization via Diversification

**Harold Benoit**[*,1,2]    **Liangze Jiang**[*,1]    **Andrei Atanov**[*,1]

**Oğuzhan Fatih Kar**[1]    **Mattia Rigotti**[2]    **Amir Zamir**[1]

[1]Swiss Federal Institute of Technology (EPFL)    [2]IBM Research

## Abstract

Supervised learning datasets may contain multiple cues that explain the training set equally well, i.e., learning any of them would lead to the correct predictions on the training data. However, many of them can be spurious, i.e., lose their predictive power under a distribution shift and consequently fail to generalize to out-of-distribution (OOD) data. Recently developed "diversification" methods (Lee et al., 2023; Pagliardini et al., 2023) approach this problem by finding multiple diverse hypotheses that rely on different features. This paper aims to study this class of methods and identify the key components contributing to their OOD generalization abilities. We show that (1) diversification methods are highly sensitive to the distribution of the unlabeled data used for diversification and can underperform significantly when away from a method-specific sweet spot. (2) Diversification alone is insufficient for OOD generalization. The choice of the used learning algorithm, e.g., the model's architecture and pretraining, is crucial. In standard experiments (classification on Waterbirds and Office-Home datasets), using the second-best choice leads to an up to 20% absolute drop in accuracy. (3) The optimal choice of learning algorithm depends on the unlabeled data and vice versa, i.e., they are co-dependent. (4) Finally, we show that, in practice, the above pitfalls cannot be alleviated by increasing the number of diverse hypotheses, the major feature of diversification methods. These findings provide a clearer understanding of the critical design factors influencing the OOD generalization abilities of diversification methods. They can guide practitioners in how to use the existing methods best and guide researchers in developing new, better ones.

## 1 Introduction

Achieving out-of-distribution (OOD) generalization is a crucial milestone for the real-world deployment of machine learning models. A core obstacle in this direction is the presence of *spurious features*, i.e., features that are predictive of the true label on the training data distribution but fail under a distribution shift. They may appear due to, for example, a bias in the data acquisition process (Oakden-Rayner et al., 2020)) or an environmental cue closely related to the true predictive feature (Beery et al., 2018).

The presence of a *spurious correlation* between spurious features and true underlying labels implies that there are multiple hypotheses (i.e., labeling functions) that all describe training data equally well, i.e., have a low training error, but only some generalize to the OOD test data. Previous works (Atanov et al., 2022; Battaglia et al., 2018; Shah et al., 2020) have shown that in the presence of multiple predictive features, standard empirical risk minimization (Vapnik, 1991) (ERM) using neural networks trained with stochastic gradient descent (SGD) converges to a hypothesis that is most aligned with the learning algorithm's inductive biases. When these inductive biases are not aligned well with the true underlying predictive feature, it can cause ERM to choose a wrong (spurious) feature and, consequently, fail under a distribution shift.

---

[*]Equal contribution. Corresponding author: harold.benoit@alumni.epfl.ch

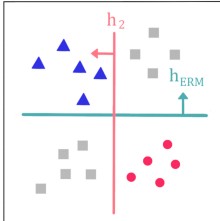 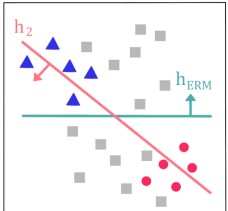 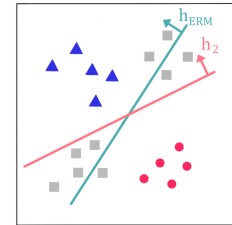

(a) Original data (b) Unlabeled data changes (c) Learning algo. changes

Figure 1: **Diversification is a *two-legged* problem where *unlabeled data* and *learning algorithm* both matter *and* are co-dependent.** ▲ and ⬤ represent the training data points and their labels. ▮ represents unlabeled data. $h_{\text{ERM}}$ represents the hypothesis found by empirical risk minimization (ERM), thus reflecting the inductive bias of the learning algorithm. $h_2$ represents a second diverse hypothesis found by a diversification method; it has low error on training data as $h_{\text{ERM}}$ does, but disagrees with it on the unlabeled data. Compared to (a) the original setting, we study how changing (b) unlabeled data and (c) the learning algorithm yield different solutions and, therefore, performance.

Recently, *diversification methods* (Lee et al., 2023; Pagliardini et al., 2023) have achieved state-of-the-art results in classification settings in the presence of spurious correlations. Instead of training a single model, these methods aim to find multiple *plausible and diverse* hypotheses that all describe the training data well, while relying on different predictive features, which is usually done by promoting different predictions on additional unlabeled data. The motivation is that among all the found features, there will be the true predictive one that is causally linked to the label and, therefore, remains predictive under a distribution shift.

In this work, we identify and study the key factors that contribute to the success of these diversification methods, adopting (Lee et al., 2023; Pagliardini et al., 2023) as two recently proposed best-performing representative methods. Our contributions are as follows.

- First, through theoretical and empirical analyses, we show that diversification methods are *sensitive to the distribution of the unlabeled data* (Fig. 1(a) vs. 1(b)). Specifically, each diversification method works best for different distributions of unlabeled data, and the performance drops significantly (up to 30% absolute accuracy) when diverging from the optimal distribution.

- Second, we demonstrate that *diversification alone cannot lead to OOD generalization efficiently without additional biases*. This is similar to the in-distribution generalization with ERM, where "good" learning algorithm's inductive biases are necessary for generalization (Vapnik & Chervonenkis, 2015). In particular, we show that *these methods are sensitive to the choice of the architecture and pretraining method* (Fig. 1(a) vs. 1(c)), and the deviation from best to second best model choice results in a significant (up to 20% absolute) accuracy drop (see Fig. 3).

- Further, we show that a *co-dependence exists between unlabeled data and the learning algorithm*, i.e., the optimal choice for one depends on the other. Specifically, for fixed training data, we can change unlabeled data in a targeted way to make one architecture (e.g., MLP) generalize and the other (e.g., ResNet18) to have random guess test accuracy and vice versa.

- Finally, we show that one of the expected advantages of diversification methods – increasing the number of diverse hypotheses to improve OOD generalization – does not hold up in practice and does not help to alleviate the aforementioned pitfalls. Specifically, we do not observe any meaningful improvements using more than two hypotheses.

These findings provide a clearer understanding of the relevant design factors influencing the OOD generalization of diversification methods. They can guide practitioners in how to best use the existing methods and guide researchers in developing new, better ones. We provide guiding principles distilled from our study in each section and Sec. 6.

## 2  RELATED WORK

**Spurious correlation and underspecification.** As a special case of OOD generalization problem, spurious correlations can arise from the underspecified nature of the training data (D'Amour et al., 2020; Koh et al., 2021; Liang & Zou, 2022). In this setting, neural networks tend to learn simple (spurious) concepts rather than the true causal concepts, a phenomenon known as simplicity bias (Shah et al., 2020; Huh et al., 2023) or shortcut learning (Geirhos et al., 2020; Scimeca et al., 2022).

Some works combat spurious correlation by improving worst-group performance (Sagawa et al., 2020; Hu et al., 2018; Zhang et al., 2021), some require group annotation (Sagawa et al., 2020; Zhang et al., 2022; Creager et al., 2021) and others (Liu et al., 2021; LaBonte et al., 2022; Sohoni et al., 2020) aim at the no group information scenario. Diversification methods fit into the latter, as they only rely on additional unlabeled data to promote diversity between multiple hypotheses.

**Diversification methods.** Recently proposed diversification methods (Lee et al., 2023; Pagliardini et al., 2023; Teney et al., 2022a;b) find multiple diverse hypotheses during training to handle spurious correlations. They introduce an additional diversification loss over multiple trained hypotheses, forcing them to rely on different features while still fitting training data well. Teney et al. (2022a;b) use *input-space* diversification that minimizes the alignment of input gradients over pairs of models at all training data points. DivDis (Lee et al., 2023) and D-BAT (Pagliardini et al., 2023) use *output-space* diversification, minimizing the agreement between models' predictions on additional unlabeled data. We focus on studying the latter, as these methods outperform the input-space ones by a large margin, achieving state-of-the-art performance in the setting where true labels are close to or completely correlated with spurious attributes.

**Inductive biases in learning algorithms.** Different learning algorithms have different inductive biases (Shalev-Shwartz & Ben-David, 2014; Hüllermeier et al., 2013), which makes a given algorithm prioritize a specific solution (Gunasekar et al., 2018; Ji & Telgarsky, 2020). While being highly overparameterized (Allen-Zhu et al., 2019) and able to fit even random labels (Zhang et al., 2017), deep learning models were shown to benefit from architectural (Xu et al., 2021; Naseer et al., 2021) , optimization (Kalimeris et al., 2019; Liu et al., 2020) and pre-training (Immer et al., 2022; Lovering et al., 2021) inductive biases. In this work, we study the influence of the choice of the learning algorithm and, hence, its inductive bias on the performance of diversification methods. We show that diversification methods are sensitive to the choice of architecture and pretraining method.

## 3 LEARNING VIA DIVERSIFICATION

First, we formalize the problem of generalization under spurious correlation. Then, we present a diversification framework along with the recent representative methods, DivDis (Lee et al., 2023) and D-BAT (Pagliardini et al., 2023)[1], describing key differences between them: training strategies (sequential vs. simultaneous) and diversification losses (mutual information vs. agreement).

### 3.1 PROBLEM FORMULATION

For consistency, we follow a notation similar to that of D-BAT (Pagliardini et al., 2023). Let $\mathcal{X}$ be the input space, $\mathcal{Y}$ the output space. Both methods focus on classification, i.e. $\mathcal{Y} = \{0, \ldots, q-1\}$, where $q$ is the number of classes. We define a domain $(D, h)$ as a distribution $D$ over $\mathcal{X}$ and a hypothesis (labeling function) $h : \mathcal{X} \to \mathcal{Y}$. The training data is drawn from the domain $(D_t, h^*)$, and test data from a different domain $(D_{\mathrm{ood}}, h^*)$. Given any domain $(D, h')$, a hypothesis $h$, and a loss function (e.g. cross-entropy loss) $\mathcal{L} : \mathcal{Y} \times \mathcal{Y} \to \mathbb{R}^+$, the expected loss is defined as: $\mathcal{L}_D(h, h') = \mathbb{E}_{x \sim D}[\mathcal{L}(h(x), h'(x))]$. Let $\mathcal{H}$ be the set of hypotheses expressed by a given learning algorithm. We define $\mathcal{H}_t^*$ and $\mathcal{H}_{\mathrm{ood}}^*$ to be the optimal hypotheses set on the train and the OOD domains:

$$\mathcal{H}_t^* := \operatorname*{argmin}_{h \in \mathcal{H}} \mathcal{L}_{D_t}(h, h^*), \quad \mathcal{H}_{\mathrm{ood}}^* := \operatorname*{argmin}_{h \in \mathcal{H}} \mathcal{L}_{D_{\mathrm{ood}}}(h, h^*). \tag{1}$$

**Definition 1.** *(**Spurious Ratio**) Given a spurious hypothesis $h$, the spurious ratio $r_D^h$, with respect to a distribution $D$ and its labeling function $h^*$ is defined as the proportion of data points where $h^*$ and $h$ agree, i.e., have the same prediction: $r_D^h = \mathbb{E}_{x \sim D}[h^*(x) = h(x)]$.*

The spurious ratio describes how the spurious hypothesis $h$ correlates with the true $h^*$ on data $D$. A spurious ratio of 1 indicates that a given data distribution has a *complete spurious correlation*. On the contrary, a spurious ratio of 0 indicates that the spurious hypothesis is always in opposition to the true labeling, namely *inversely correlated*. Finally, a spurious ratio of 0.5 means that the spurious hypothesis is not predictive of $h^*$, as there is no correlation between them. We will also refer to this setting as a *"balanced"* data distribution. We omit $h^*$ (and sometimes $h$) in the notation to keep it less cluttered, as they can be inferred from the context of a specific setting.

---

[1]At the time of writing, these are the best-performing diversification methods and the only existing output-space ones.

**Spurious correlation setting.** In this setting, we assume that there exist one or more spurious hypotheses $h \in \mathcal{H}_{\mathrm{sp}} \subset \mathcal{H}_t^* \setminus \mathcal{H}_{\mathrm{ood}}^*$, which generalize on $D_t$ but not on $D_{\mathrm{ood}}$. Thus, the spurious ratio on training data is close to one: $r_{D_t}^h \approx 1$. If there is a misalignment between the inductive bias of the learning algorithm and $\mathcal{H}_{\mathrm{ood}}^*$, the ERM hypothesis $h_{\mathrm{ERM}}$ may be closer to hypotheses from $\mathcal{H}_{\mathrm{sp}}$ than $\mathcal{H}_t^* \cap \mathcal{H}_{\mathrm{ood}}^*$, i.e., have poor OOD generalization. The idea of diversification methods is to find multiple hypotheses from $\mathcal{H}_t^*$ with the aim to have one with good OOD generalization (see Sec. 3.2).

## 3.2 DIVERSIFICATION FOR OOD GENERALIZATION

DivDis and D-BAT focus on the above spurious correlation setting. They assume access to additional unlabeled data $D_u$ to find multiple diverse hypotheses that all fit the training data $(D_t, h^*)$ but disagree, i.e., make diverse predictions, on $D_u$. The motivation is to better cover the space $\mathcal{H}_t^*$ and, consequently, find a hypothesis from $\mathcal{H}_t^* \cap \mathcal{H}_{\mathrm{ood}}^*$ that also generalizes to OOD data.

**Optimization objective.** Following DivDis and D-BAT, we define a diversification loss $A_{D_u}(h_1, h_2)$ that quantifies the agreement between two hypotheses on $D_u$. Then, in the case of finding $K$ hypotheses, the training objective of a diversification method is the sum of ERM loss and the diversification loss averaged over all pairs of hypotheses:

$$h_1, ..., h_K = \underset{h_1,...,h_K \in \mathcal{H}}{\operatorname{argmin}} \sum_{i=1}^{K} \mathcal{L}_{D_t}(h_i, h^*) + \frac{\alpha}{K(K-1)} \sum_{i=1}^{K} \sum_{j=1, \ i \neq j}^{K} A_{D_u}(h_i, h_j), \qquad (2)$$

**Diversification loss.** Let $P_{h_i}$ be the predictive distribution of a hypothesis $h_i$ on given data $D$. Then,

- **DivDis** (Lee et al., 2023): $A_D(h_1, h_2) = D_{\mathrm{KL}}(P_{(h_1,h_2)} || P_{h_1} \otimes P_{h_2}) + \lambda \sum_{i \in \{1,2\}} D_{\mathrm{KL}}(P_{h_i} || \hat{P})$. The first term is the mutual information, which is equal to 0 iff $P_{h_1}$ and $P_{h_2}$ are independent. The second term is the KL-divergence between $P_{h_i}$ and a prior distribution $\hat{P}$ (usually set to the distribution of labels in $D_t$). It prevents hypotheses from collapsing to degenerate solutions.
- **D-BAT** (Pagliardini et al., 2023): $A_D(h_1, h_2) = \mathbb{E}_{x \sim D}[-\log(P_{h_1}(x;0) \cdot P_{h_2}(x;1) + P_{h_1}(x;1) \cdot P_{h_2}(x;0))]$, where $P_{h_i}(x;y)$ is the probability of class $y$ predicted by $h_i$.

In practice, they are computed and optimized on additional unlabeled data $D_u$. Note that it is usually favorable to have the distribution of $D_u$ different from that of $D_t$, i.e., $r_{D_u}^h < r_{D_t}^h \approx 1$, as this enables the diversification process to distinguish between spurious and semantic hypotheses (This is also confirmed by empirical results in Fig.2-Right). In Sec. 4, we will show that both losses have their strengths, and the optimal choice depends on the spurious ratio of $D_u$.

**Sequential vs. simultaneous optimization.** In practice, when minimizing the diversification objective in Eq. 2, there are two choices: (i) optimize over all hypotheses simultaneously or (ii) find hypotheses one by one. DivDis trains simultaneously and defines hypotheses as linear classifiers that share the same feature extractor. D-BAT, on the contrary, starts with $h_1 \triangleq h_{\mathrm{ERM}}$ and finds new hypotheses, defined as separate models sequentially. *For consistency and comparability with D-BAT in Sec. 4 analysis, we also introduce **DivDis-Seq**, a version of DivDis using sequential optimization*, allowing us to concentrate only on the difference in diversification loss design.

**The two-stage framework.** After finding $K$ hypotheses, one needs to be chosen to make the final prediction, leading to a two-stage approach (Lee et al., 2023): 1. *Diversification*: find $K$ diverse hypotheses $\mathcal{H}_K^* \subset \mathcal{H}_t^*$. 2. *Disambiguation*: select one hypothesis $\hat{h} \in \mathcal{H}_K^*$ given additional information (e.g., a few labeled test examples or human supervision). We identify the diversification stage as *the most critical* one. Indeed, if the desired hypothesis $h^*$ is not chosen ($h^* \notin \mathcal{H}_K^*$), the second stage cannot make up for it as it is limited to only hypotheses from $\mathcal{H}_K^*$. We, therefore, focus on the first stage and assume an oracle that chooses the best available hypothesis in the second stage.

## 4 THE RELATIONSHIP BETWEEN UNLABELED DATA AND OOD GENERALIZATION VIA DIVERSIFICATION

In this section, we study how the different diversification losses of DivDis and D-BAT interact with the choice of unlabeled data. In an illustrative example and real-world datasets, we identify that neither of the diversification losses is optimal in all scenarios and that their behavior and performance are highly dependent on the spurious ratio of the unlabeled OOD data $D_u$.

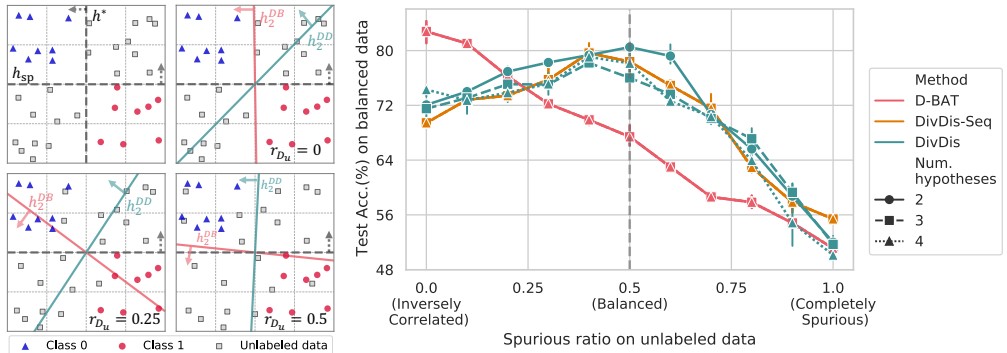

Figure 2: **Performance of diversification is *highly dependent* on unlabeled OOD data. Left:** *Top-left quadrant*: The 2D binary classification task. *Other quadrants*: Show the second hypotheses (arrows are normal vectors) found by D-BAT ($h_2^{\mathrm{DB}}$) and DivDis-Seq ($h_2^{\mathrm{DD}}$) with varied spurious ratios of unlabeled OOD data $r_{D_u} = \{0, 0.25, 0.5\}$ (from inversely correlated to balanced). **Right:** Best hypothesis test accuracy of D-BAT & DivDis(-Seq) on MNIST-CIFAR (M/C) for varied spurious ratios $r_{D_u}$ and number of hypotheses $K$. The test accuracy is measured on hold-out balanced data $D_{\mathrm{ood}}$ (i.e., $r_{D_{\mathrm{ood}}} = 0.5$, no spurious correlation).

## 4.1 THEORETICAL AND EMPIRICAL STUDY OF A SYNTHETIC EXAMPLE

**Synthetic 2D binary classification task.** In Fig. 2-Left, we show a 2D task with distribution $D_{\mathrm{ood}}$ spanning a 2D square, i.e., $\{x = (x_1, x_2) \in [-1, 1]^2\}$. We define our hypothesis space $\mathcal{H}$ to be all possible linear classifiers $h(x; \beta)$ where $\beta$ is the radian of the classification plane w.r.t horizontal axis $x_1$. The ground truth labeling function is defined as $h^\star(x) = h(x; \frac{\pi}{2}) = \mathcal{I}\{x_1 > 0\}$ where $\mathcal{I}$ is the indicator function, and the training distribution is defined as $D_t = \{x = (x_1, x_2) \in \{[-1, 0] \times [0, 1]\} \cup \{[0, 1] \times [-1, 0]\}\}$. We then define a spurious feature function as $h_{\mathrm{sp}}(x) = h(x; 0) = \mathcal{I}\{x_2 < 0\}$ and assume that ERM converges to $h_{\mathrm{sp}}$. This means that the first hypothesis $h_1^{DB}$ (D-BAT) and $h_1^{DD}$ (DivDis-Seq) of both methods converge to $h_{\mathrm{sp}}$. Finally, we define different distributions of unlabeled data $D_u$ to have different spurious ratios $r_{D_u}$ from 0 to 0.5 (the construction is described in Appendix A).

**Proposition 1.** *(On Optimal Diversification Loss) In the synthetic 2D binary task, let $h_2^{DB}$ and $h_2^{DD}$ be the second hypotheses of D-BAT and DivDis-Seq, respectively. If $r_{D_u} = 0$, then $h_2^{DB} = h^\star$ and $h_2^{DD} = h(x; \frac{\pi}{4})$. Otherwise, if $r_{D_u} = 0.5$, then $h_2^{DB} = h(x; \pi) = 1 - h_{\mathrm{sp}}$ and $h_2^{DD} = h^\star$. Increasing the spurious ratio $r_{D_u}$ from 0 to 0.5 will lead to $h_2^{DB}$ and $h_2^{DD}$ rotating counterclockwise.*

See Appendix A for the full proof and Fig. 2-Left for the visual demonstration. This proposition implies that D-BAT recovers $h^\star$ when $r_{D_u} = 0$ (i.e., inversely correlated) and DivDis-Seq recovers $h^\star$ when $r_{D_u} = 0.5$ (i.e., balanced). For D-BAT, this happens because the optimal second hypothesis $h_2^{DB}$ is the hypothesis that disagrees with $h_1^{DB}$ on all unlabeled data points i.e. $h_2^{DB} \in \{h \in \mathcal{H}_t^* : h(x) \neq h_1^{DB}(x) \ \forall x \in D_u\}$. On the contrary, the optimal second hypothesis for DivDis-Seq is independent of the first one, i.e., disagreeing on half of the data points $h_2^{DD} \in \{h \in \mathcal{H}_t^* : \mathbb{P}_{x \sim D_u}[h(x) = h_1^{DD}(x)] = \frac{1}{2}\}$.

In Fig. 2-Left, we empirically demonstrate this behavior by training linear classifiers[2] with D-BAT and DivDis-Seq[3] on such synthetic data, with 0.5k training / 5k unlabeled OOD data points (following (Lee et al., 2023)). We observe that the behavior suggested in Proposition 1 is consistent with our experiments. This highlights that different diversification losses only recover the ground truth function in different specific spurious ratios.

## 4.2 VERIFICATION ON REAL-WORLD IMAGE DATA

We further evaluate whether the suggested behavior holds with more complex classifiers and more complex datasets. Specifically, we use M/C (Shah et al., 2020) and M/F (Pagliardini et al., 2023),

---

[2]In Appendix B, we show additional results with more complex classifiers (i.e., MLP).

[3]For completeness, we also provide results with DivDis, which is deferred to Appendix C.

which are datasets that concatenate one image from MNIST with one image from either CIFAR-10 (Krizhevsky & Hinton, 2009) or Fashion-MNIST (Xiao et al., 2017). We follow the setup of Lee et al. (2023); Pagliardini et al. (2023): we use 0s and 1s from MNIST and two classes from Fashion-MNIST (coats & dresses) and CIFAR-10 (cars & trucks). The training data is designed to be completely spuriously correlated (e.g., 0s always occur with cars and 1s with trucks in M/C). We vary the spurious ratio $r_{D_u}$ of the unlabeled data by changing the probability of 0s occurring with cars/dresses. We use LeNet (Lecun et al., 1998) architecture for both D-BAT and DivDis(-Seq) methods.

Fig. 2-Right shows that similar to Proposition 1, D-BAT is optimal when $r_{D_u} = 0$ whereas DivDis(-Seq) optimal setting is $r_{D_u} = 0.5$. Both methods observe a drastic decrease in performance away from their "sweet spot" (with up to 30% absolute accuracy drop). Note that it is expected that both methods reach chance-level accuracy when $r_{D_u} \to 1$, as it means that the spurious hypothesis becomes completely correlated to the true hypothesis on $D_u$, and it is thus impossible to differentiate them by enforcing diversification on $D_u$. In Appendix D, Fig. 9 shows that the same observation holds for the M/F dataset, and Tab. 4 also shows the results of different spurious ratios on a larger and more realistic dataset, CelebA-CC (Liu et al., 2015; Lee et al., 2023).

Lee et al. (2023); Pagliardini et al. (2023) note that both the number of hypotheses $K$ and the diversification coefficient $\alpha$ (Eq. 2) are critical hyper-parameters, that may greatly influence the performance. However, controlling for these variables, in Fig. 2-Right and Fig. 10, we find that tuning $\alpha$ and $K$ is not sufficient to compensate the performance loss from the misalignment between unlabeled OOD data and the diversification loss.

> **Takeaway.** Diversification methods' performance drops drastically when away from the spurious ratio (Def. 1) "sweet spot", and neither diversification loss is optimal in all cases. Therefore, new methods should be designed to adapt to different unlabeled data distributions.

## 5 THE RELATIONSHIP BETWEEN LEARNING ALGORITHM AND OOD GENERALIZATION VIA DIVERSIFICATION

In this section, we study another key component of diversification methods – the choice of the learning algorithm. First, we present a theoretical result showing that diversification alone is insufficient to achieve OOD generalization and requires additional biases (e.g., the inductive biases of the learning algorithm). Then, we empirically demonstrate the high sensitivity of these methods to the choice of the learning algorithm (architecture and pretraining method). Finally, empirically, we show that the optimal choices of the learning algorithm and unlabeled data are co-dependent.

### 5.1 DIVERSIFICATION ALONE IS INSUFFICIENT FOR OOD GENERALIZATION

Diversification methods find hypotheses $h_i$s that all minimize the training loss, i.e., $h_i \in \mathcal{H}_t^*$, but disagree on the unlabeled data $D_u$. The underlying idea is to cover the space $\mathcal{H}_t^*$ evenly and better approximate a generalizable hypothesis from $\mathcal{H}_t^* \cap \mathcal{H}_{\text{ood}}^*$ (e.g., see Fig. 3 in (Pagliardini et al., 2023)). However, if the original hypothesis space $\mathcal{H}$ is expressive enough to include all possible labeling functions (e.g., neural networks (Hornik et al., 1989)), then $\mathcal{H}_t^*$ essentially only constrains its hypotheses' labeling on the training data $D_t$ while including all possible labelings over $D_{\text{ood}}$, which implies $q^{|D_{\text{ood}}|}$ possible labelings, where $q$ is the number of classes. Therefore, one might need to find exponentially many hypotheses before covering this space and approximating the desired hypothesis $h^* \in \mathcal{H}_t^* \cap \mathcal{H}_{\text{ood}}^*$ well enough.

Notably, we prove that having as many diverse hypotheses as the number of data points in $D_{\text{ood}}$ is still insufficient to guarantee better than a random guess accuracy. Indeed, there always exists a set of hypotheses satisfying all the constraints of the diversification objective in Eq. 2 while having random accuracy w.r.t the true labeling $h^*$ on OOD data. The following Proposition 2 formalizes this intuition in the binary classification case. Please see its proof and extension to the multi-class case in Appendix E.

**Proposition 2.** *For $K = (2|D_{\text{ood}}| - 1)$ and $h^*$ the OOD labeling function, there exists a set of diverse $K$ hypotheses $h_1, ..., h_K$, i.e., $A_{D_{\text{ood}}}(h_i, h_j) = |\{x \in D_{\text{ood}} : h_i(x) = h_j(x)\}|/|D_{\text{ood}}| \leq 0.5 \quad \forall i, j \in \{1, ..., K\}, i \neq j$ and it holds that $\max_{h' \in h_1, ..., h_K} \text{Acc}(h^*, h') \leq 0.5$.*

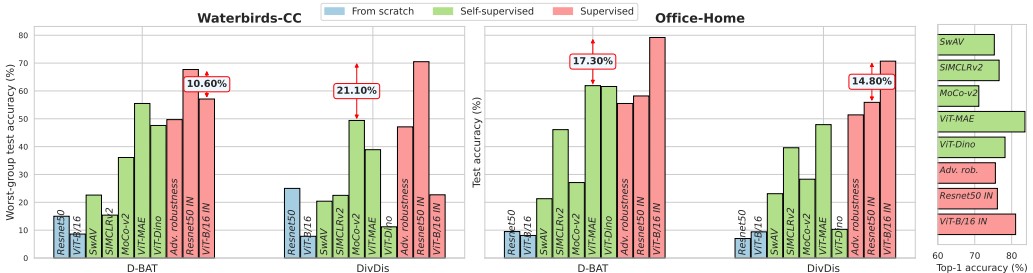

Figure 3: **The performance of diversification methods is highly sensitive to the choice of architecture and pretraining method. Left:** DivDis and D-BAT best hypothesis ($K = 2$) performance with multiple pretraining strategies and architecture pairs on Waterbirds-CC (Left) and Office-Home (Right). ResNet50 is used if not specified. **Right:** Top-1 accuracy on ImageNet-1k after fine-tuning.

Since in most cases, Lee et al. (2023); Pagliardini et al. (2023) find 2 hypotheses to be sufficient to approximate $h^*$ and the size of the used datasets is larger than 2, these hypotheses should not only be diverse but also *biased* towards those that generalize under the considered distribution shift.

> **Takeaway.** Diversification alone cannot lead to OOD generalization efficiently and requires additional biases to be brought by a specific learning algorithm used in practice.

**Properties of diverse hypotheses.** In Appendix F.2, using the agreement score (AS) (Atanov et al., 2022; Hacohen et al., 2020) as a measure of the alignment of a hypothesis with a learning algorithm's inductive biases, we study in what way D-BAT and DivDis diverse hypotheses are biased. We show that they find hypotheses that are not only diverse but aligned with the inductive bias of the used learning algorithm. According to the definition of AS, such alignment is expected for a hypothesis found by empirical risk minimization (ERM). However, it is not generally expected from diverse hypotheses (as defined in Eq. 2), given that the additional diversification loss could destroy this alignment. This analysis sheds light on the process by which diverse hypotheses are found and emphasizes the choice of a good learning algorithm, which is crucial, as shown in the next section.

## 5.2 LEARNING ALGORITHM SELECTION: A KEY TO EFFECTIVE DIVERSIFICATION

Sec. 5.1 argues that the right learning algorithm's inductive biases (i.e., those aligned well with the true causal hypothesis $h^*$) are required for diversification to enable OOD generalization. In this section, we examine the "sensitivity" of this requirement by using DivDis and D-BAT with different choices of pretraining strategies and architectures on several real-world datasets.

**Experimental setup.** We consider the following datasets. (1) A multi-class classification dataset Office-Home (Venkateswara et al., 2017) consists of images of 65 item categories across four domains: Art, Product, Clipart, and Real-World. Following the experimental setting in Pagliardini et al. (2023), we use the Product and Clipart domains during training and the Real-World domain as the out-of-distribution one. (2) A binary classification dataset: Waterbirds-CC Lee et al. (2023); Sagawa et al. (2020), a modified version of Waterbirds where the background and bird features are completely spuriously correlated on the training data. We report worst-group accuracy for Waterbirds-CC, i.e., the minimum accuracy among the four possible groups. We train both diversification methods using different architectures and pretraining methods, each resulting in a different learning algorithm with different inductive biases. Please see full experimental details and results in Appendix G.

**Sensitivity to the model choice.** Fig. 3 shows that the performance of both diversification methods is highly sensitive to the choice of the learning algorithm: 1) the gap between the best and second-best model is significant (10%-20%) 2) there is no single model that performs the best over both datasets, and 3) there is a 20% standard deviation of the performance over the distribution of models (averaged over methods and datasets). Furthermore, similar to the findings of Wenzel et al. (2022), one cannot choose a good model reliably based on the ImageNet performance as a proxy. Indeed, the best model, according to this proxy, ViT-MAE (He et al., 2021), underperforms significantly in all cases. Additionally, ViT-Dino (Caron et al., 2021), the third best on ImageNet, completely fails for DivDis

| Dataset | Method | $K=2$ | $K=3$ | $K=4$ | $K=5$ |
|---------|--------|-------|-------|-------|-------|
| Waterbirds-CC | D-BAT (ViT-B/16 IN) | $57.1_{\pm 3.7}$ | $57.1_{\pm 3.7}$ | $57.1_{\pm 3.7}$ | $57.1_{\pm 3.7}$ |
| | DivDis (MoCo-v2) | $49.4_{\pm 10.3}$ | $51.7_{\pm 6.0}$ | $49.6_{\pm 8.3}$ | $48.4_{\pm 0.9}$ |
| Office-Home | D-BAT (ViT-MAE) | $61.9_{\pm 0.7}$ | $62.6_{\pm 0.1}$ | $62.6_{\pm 0.1}$ | $62.6_{\pm 0.1}$ |
| | DivDis (Resnet50 IN) | $55.9_{\pm 0.6}$ | $54.6_{\pm 0.1}$ | $53.6_{\pm 0.4}$ | $53.1_{\pm 0.2}$ |

Table 1: **Increasing the number of hypotheses does not bridge the performance gap between different models.** We increase the number of hypotheses found by diversification methods for the second-best model in Fig. 3 and find that it is not enough to bridge the performance gap with the best-performing model. The best hypothesis accuracy is reported. Results are averaged over 3 seeds. Standard deviations for Waterbirds-CC are larger due to the usage of the worst-group accuracy metric.

on both datasets. Overall, these results emphasize the need for a *specific* architecture and pretraining tailored for each dataset and method, which may require an expensive search.

**Increasing $K$ does not improve performance.** Finally, we study whether the performance gap between the best and second-best models tested in Fig. 3 can be closed by increasing the number of hypotheses $K$, as this is allegedly the major feature and motivation of diversification methods. Tab. 1 shows that similar to the observation made in Sec. 4.2, increasing $K$ does not bring any improvements, suggesting that the choice of the model is more important for enabling OOD generalization. In Fig. 14, we further show that DivDis does not scale well to larger $K$ (e.g., $K = 64$) "out-of-the-box", and the performance drops as the number of hypotheses increases. Note that testing D-BAT in this regime would be prohibitively expensive.

> **Takeaway.** Diversification methods are highly sensitive to the choice of the learning algorithm, e.g., architecture and pretraining method. The "built-in" mechanism of increasing the number of hypotheses $K$ does not alleviate this issue and fails to improve performance.

### 5.3 On the Co-Dependence between Learning Algorithm and Unlabeled Data

Sec. 4 and Sec. 5.2 show the sensitivity of the diversification methods to the distribution of the unlabeled data and the choice of a learning algorithm, respectively. Here, we further demonstrate that these choices are *co-dependent*, i.e., the optimal choice for one depends on the other. Specifically, we show that by only varying the distribution of unlabeled data, the optimal architecture can be changed.

**Experimental setup.** We consider two learning algorithms (architectures) $\mathcal{A} \in \{\text{MLP}, \text{ResNet18}\}$ (extension to other architectures is straightforward) and construct examples for D-BAT where one architecture outperforms the other and vice versa. To do that we build on the idea of adversarial splits introduced in (Atanov et al., 2022), defined on a CIFAR-10 Krizhevsky & Hinton (2009) dataset $D$. Below, we briefly describe the construction and refer the reader to Appendix H for more details.

We start by considering two hypotheses with high agreement scores (Baek et al., 2022) found by Atanov et al. (2022) for each architecture, such that the following holds:

$$\text{AS}_{\text{MLP}}(h_{\text{MLP}}) > \text{AS}_{\text{MLP}}(h_{\text{RN}}), \quad \text{AS}_{\text{RN}}(h_{\text{RN}}) > \text{AS}_{\text{RN}}(h_{\text{MLP}}), \qquad (3)$$

where $\text{AS}_{\mathcal{A}}$ stands for the agreement score measured with algorithm $\mathcal{A}$. As shown in Atanov et al. (2022), the above inequalities suggest that each hypothesis $h_{\mathcal{A}}$ is more aligned with its corresponding learning algorithm $\mathcal{A}$, i.e., ERM trained with MLP architecture will preferentially converge to $h_{\text{MLP}}$ over $h_{\text{RN}}$ and vice-versa when training with Resnet18. Akin to adversarial splits (Atanov et al., 2022), we then use these two high-AS hypotheses to construct a dataset to change, in a *targeted* way, what the first hypothesis of D-BAT $h_1 \triangleq h_{\text{ERM}}$ converges to, depending on the used learning algorithm. Different $h_1$s, in turn, lead to different $h_2$s and, hence, different test performance.

As the true labeling $h^*$, we use a binary classification task constructed by splitting the original 10 classes into two sets of five. Then, as Tab. 2-Right illustrates, we construct training data $D_t$ to contain samples where all $h^*$, $h_{\text{MLP}}$, and $h_{\text{RN}}$ agree, i.e., $D_t = \{x \in D : h^*(x) = h_{\text{MLP}}(x) = h_{\text{RN}}(x)\}$. Thus, by design, both $h_{\text{MLP}}$ and $h_{\text{RN}}$ are completely spuriously correlated with $h^*$. Then, we define unlabeled OOD data $D_u$ s.t. either $(r_{D_u}^{h_{\text{MLP}}} = 0, r_{D_u}^{h_{\text{RN}}} = \frac{1}{2})$ (denoted as $h^* \perp h_{\text{RN}}$), or

| $D_u$ | | $\mathcal{A}$ | Test Acc.(%) |
|---|---|---|---|
| $h^\star \perp h_{\mathrm{MLP}}$ | ① | MLP | $89.2_{\pm 0.8}$ |
| | ② | ResNet18 | $56.7_{\pm 0.8}$ |
| $h^\star \perp h_{\mathrm{RN}}$ | ③ | MLP | $55.4_{\pm 0.3}$ |
| | ④ | ResNet18 | $77.0_{\pm 0.7}$ |

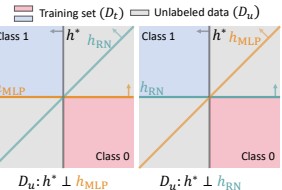

Table 2: **The optimal choices of the (learning algorithm) architecture and unlabeled data are co-dependent. Left**: Performance of the D-BAT method using two architectures and two created unlabeled data distributions. The test accuracy is on hold-out data $D_{\mathrm{ood}} \sim D_u$. **Right**: Illustration of the two unlabeled data distributions used in the experiment. We keep the training data fixed and change $D_u$ s.t. $h^\star$ is inversely correlated (denoted by '$\perp$') to one of the "spurious" hypotheses $h_{\mathrm{MLP}}, h_{\mathrm{RN}}$. *We show that the optimal choice of the architecture ($\mathcal{A}$) depends on the unlabeled data distribution*, and the best $\mathcal{A}$ is the one for which $h^* \perp h_{\mathcal{A}}$ on $D_u$.

$(r^{h_{\mathrm{MLP}}}_{D_u} = \frac{1}{2}, r^{h_{\mathrm{RN}}}_{D_u} = 0)$ (denoted as $h^* \perp h_{\mathrm{MLP}}$). This means that $h^*$ is inversely correlated with only one of $h_{\mathrm{MLP}}$ or $h_{\mathrm{RN}}$, while not correlated to the other hypothesis.

**Results.** Keeping the training data fixed, we train D-BAT ($K = 2$) using different architecture and construct unlabeled data pairs $(\mathcal{A}, D_u)$. Tab. 2-Left shows that the performance of $\mathcal{A}$ drops to almost random chance when $h_{\mathcal{A}}$ does not inversely correlate with $h^*$ on the unlabelled data (② and ③). This is consistent with Sec. 4, where we show that the setting with $r^{h_1}_{D_u} = \frac{1}{2}$ is disadvantageous for D-BAT. In Appendix H, Tab. 11 further shows similar observation for a different architecture pair (ViT & ResNet18), and Fig. 15 extends the experiment with smooth interpolation from one unlabeled dataset setting to the other, showing a linear transition where one architecture goes from optimal performance to random-chance accuracy, and vice-versa.

> **Takeaway.** The optimal choices of the architecture and unlabeled data are co-dependent.

## 6 CONCLUSION AND LIMITATIONS

This paper aims to study diversification methods and identify key components enabling their OOD generalization: the diversification loss used, the distribution of the unlabeled data, and the choice of a learning algorithm. Below, we distill some practical recommendations that follow from our analysis.

**Unlabeled data and diversification loss.** Sec. 4 shows that a sub-optimal spurious ratio w.r.t to the chosen diversification loss may lead to significant performance drops. One possibility to overcome this problem is to use a mixture of diversification losses, determined by an estimate of the spurious ratio of unlabeled data. Another is to try to collect unlabeled data with a specific spurious ratio.

**Choice of the learning algorithm.** Sec. 5.2 demonstrates that the methods are highly sensitive to the choice of the learning algorithm inductive bias. Future methods should be made more resilient to this choice, e.g., by modeling each hypothesis with different architectures and pretraining methods or by implementing a mechanism to choose a "good" model automatically.

**Co-dependence.** Sec. 5.3 suggests that a practitioner should not expect the best learning algorithm (e.g., architecture or pretraining choice) found on one dataset to perform well on another one (as observed in Sec. 5.2), and an additional search might be needed to achieve good performance.

Then we discuss the limitations of our study:

**Data characteristics.** We characterize the influence of the OOD data distribution through its spurious ratio. The influence of other important properties of the OOD data may need to be studied in future work. Furthermore, we mainly focused on image data to aid the comparison with Lee et al. (2023); Pagliardini et al. (2023), but we expect our conclusions to be mainly data-agnostic.

**Co-dependence experiment only with D-BAT** In Sec. 5.3, the experiment is only performed with D-BAT. We expect DivDis to have a similar co-dependence. However, its diversification loss (mutual information) and optimization strategy (simultaneous) make such a targeted experiment challenging to design. We leave an explicit demonstration for future work.

## REPRODUCIBILITY STATEMENT

In order to ensure that this work is reproducible, we have taken the following steps. In Appendix A and E, we provide proofs for each theoretical result (Proposition 1 and Proposition 2). For the experiments, in Appendix D, G, and H, we provide a complete description of the datasets, used models, and hyper-parameter settings. Additionally, all results from DivDis (Lee et al., 2023) and D-BAT (Pagliardini et al., 2023) are obtained using their respective published source code, ensuring a faithful representation of their methods. Finally, we provide the anonymized source code for the experiments performed in the paper.

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

# Appendix

The appendix of this work is outlined as follows:

- Appendix A proves Proposition 1 of Sec. 4.1 (synthetic 2D task), and **shows that the optimal diversification loss depends on the spurious ratio of the unlabeled data**.

- Appendix B extends the experiment done in Sec. 4.1 (synthetic 2D task) by training a multilayer perceptron (MLP) instead of a linear classifier, and **shows empirically that Proposition 1 extends to more complex classifiers.**

- Appendix C provides additional experiments for Sec. 4.1, and **shows empirically that Proposition 1 extends to DivDis**.

- Appendix D provides the implementation details of the experimental verification of Proposition 1 on real-world images (Sec. 4.2) **We also provide additional results, using the M/F dataset** (where MNIST and Fashion-MNIST (Xiao et al., 2017) are concatenated), **as well as the CelebA (Liu et al., 2015) dataset**. We also show that **tuning the diversification hyperparameter $\alpha$ is *not* sufficient to compensate the performance loss from the misalignment between unlabeled data and diversification loss**, i.e., the conclusion of Proposition 1 still holds when tuning $\alpha$.

- Appendix E proves Proposition 2 of Sec. 5.1, **proving the existence of a large number of pairwise diverse hypotheses which do not generalize**. A proof for a similar result in the multi-class classification case is also provided.

- Appendix F.1 provides **an overview of the important concepts from Task Discovery** (Atanov et al., 2022) used in this paper (agreement score, adversarial splits).

- Appendix F.2, using agreement score, explains the experimental setup and results that demonstrate that **D-BAT and DivDis find hypotheses that are not only diverse but aligned with the inductive bias of the used learning algorithm.**

- Appendix G reports the experimental details and full results of Sec. 5.2.

- Appendix H provides a detailed explanation of how to construct the training and unlabeled data of 5.3 where **we show that by only changing the distribution of unlabeled data, we can influence the optimal choice of the architecture.** It also contains a variant of Tab. 2 with ViT&ResNet pair, as well as an extension of the experiment **with a smooth interpolation from one unlabeled dataset setting to the other, showing a linear transition where one architecture goes from the optimal performance to random-chance accuracy, and vice-versa.**

## A    PROOF AND DISCUSSION OF PROPOSITION 1

In Sec. 4, we make a proposition that, in the synthetic 2D example, the optimal choice of diversification loss changes with the spurious ratio of unlabeled OOD data $r_{D_u}$. Specifically, DivDis-Seq finds the ground truth hypothesis $h^\star$ if and only if $r_{D_u} = 0.5$ (i.e., balanced or no spurious correlation), whereas D-BAT discovers $h^\star$ if and only if $r_{D_u} = 0$ (i.e., inversely correlated). In this section, we provide the proof, method by method, and case by case.

We first restate the Proposition 1 as follows:

**Synthetic 2D Binary Classification Task.** We illustrate the setting in Fig. 4 and describe it below:

- The data domain spans a 2D square, i.e., $\{x = (x_1, x_2) \in [-1, 1]^2\}$.
- The training distribution is defined as $D_t = \{x = (x_1, x_2) \in \{[-1, 0] \cup [0, 1]\} \cup \{[0, 1] \cup [-1, 0]\}\}$, i.e., contains data points the 1st and 4th quadrants.
- Our hypothesis space $\mathcal{H}$ contains all possible linear classifiers $h(x; \beta)$ where $\beta$ is the radian of the classification plane w.r.t horizontal axis $x_1$.
- The ground truth hypothesis is $h^\star(x) = h(x; \frac{\pi}{2}) = \mathcal{I}\{x_1 > 0\}$, where $\mathcal{I}$ is the indicator function.

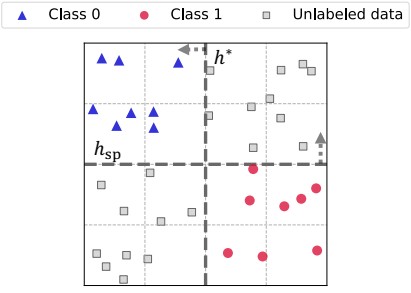

Figure 4: **Synthetic 2D Binary Classification Task.** ▲ and ● represent the training data points and their labels. ▪ represents unlabeled OOD data. In this setting, the unlabeled OOD data $D_u$ has spurious ratio $r_{D_u} = 0$ (i.e., inversely correlated).

- The spurious hypothesis, i.e. the one that ERM converges to, is assumed to be $h_{\text{sp}}(x) = h(x; 0) = \mathcal{I}\{x_2 < 0\}$.
- Thus, $h_{\text{sp}}$ and $h^\star$ agree on the training data (1st and 4th quadrants) and disagree on the 2nd and 3rd quadrants.
- We vary the spurious ratio of the unlabeled OOD data distribution $D_u$ by varying the ratio of data points sampled from the 1st and 4th quadrants over the number of data points sampled from the 2nd and 3rd quadrants.
- One possibility is to define $D_u = \{x = (x_1, x_2) \in \{[R(r_{D_u}), 1] \cup [0, 1]\} \cup \{[-1, -R(r_{D_u})] \cup [-1, 0]\}\}$, and $R(r) = \frac{r}{r-1}$ for $0 \leq r \leq 0.5$.
- Let $P_h(x; y)$ be the probability of class $y$ predicted by hypothesis $h$ given sample $x$. The following proof assumes both the hypotheses $h_{\text{sp}}$ and the second hypothesis $h_2^{DB}$ or $h_2^{DD}$ discovered by D-BAT and DivDis-Seq have a hard margin, i.e., $P_h(x; y) \in \{0, 1\}$. Nonetheless, we also show empirically in Sec. 4.1 (Fig. 2) that when this hard margin condition does not hold, we get the same conclusion as Proposition 1.

**Proposition 1.** *(On Optimal Diversification Loss) In the synthetic 2D binary task, let $h_2^{DB}$ and $h_2^{DD}$ be the second hypotheses of D-BAT and DivDis-Seq, respectively. If $r_{D_u} = 0$, then $h_2^{DB} = h^\star$ and $h_2^{DD} = h(x; \frac{\pi}{4})$. Otherwise, if $r_{D_u} = 0.5$, then $h_2^{DB} = h(x; \pi) = 1 - h_{\text{sp}}$ and $h_2^{DD} = h^\star$.*

*Proof.*

In the following proof, we use $-h$ to denote the opposite hypothesis, i.e., $-h(x) = 1 - h(x)$.

**D-BAT.** Plugging in the unlabeled OOD data distribution $D_u$, the first hypothesis $h_{\text{sp}}$ and the second hypothesis $h_2^{DB}$, the diversification loss in D-BAT is:

$$A_{D_u}(h_{\text{sp}}, h_2^{DB}) = \mathbb{E}_{x \sim D_u}[-\log(P_{h_{\text{sp}}}(x; 0) \cdot P_{h_2^{DB}}(x; 1) + P_{h_{\text{sp}}}(x; 1) \cdot P_{h_2^{DB}}(x; 0))],$$

Let $\mathcal{L}_{D_t}^{\text{ERM}}$ be the ERM loss on fitting training data. D-BAT's objective is then $\mathcal{L}_{D_t}^{\text{ERM}} + \alpha A_{D_u}$, where $\alpha$ is a hyperparameter. Bellow, we prove the proposition case by case:

- When $r_{D_u} = 0$ (inversely correlated), the unlabeled OOD data spans $\{[0, 1] \cup [0, 1]\} \cup \{[-1, 0] \cup [-1, 0]\}$, i.e., the second and third quadrants. In this case, the diversification loss is:

$$A_{D_u} = \mathbb{E}_{x \sim D_u}[-\log(P_{h_{\text{sp}}}(x; 0) \cdot P_{h_2^{DB}}(x; 1) + P_{h_{\text{sp}}}(x; 1) \cdot P_{h_2^{DB}}(x; 0))]$$

$$= \mathbb{E}_{x \sim D_u}[-\log(1 \cdot P_{h_2^{DB}}(x; 1) + 0 \cdot P_{h_2^{DB}}(x; 0)) \mid x \in \{[0, 1] \cup [0, 1]\}] \cdot \frac{1}{2} \quad \text{(2nd quadrant)}$$

$$+ \mathbb{E}_{x \sim D_u}[-\log(0 \cdot P_{h_2^{DB}}(x; 1) + 1 \cdot P_{h_2^{DB}}(x; 0)) \mid x \in \{[-1, 0] \cup [-1, 0]\}] \cdot \frac{1}{2} \quad \text{(3rd quadrant)},$$

$$\tag{4}$$

where we assume uniform distribution over $D_{\text{ood}}$. The hypothesis $h_2^{DB}$ which minimizes the diversification loss in Eq. 4 should satisfy $P_{h_2^{DB}}(x;1) = 1$ and $P_{h_2^{DB}}(x;0) = 1$ for the data points in the second and third quadrants, respectively. Since the hypothesis space consists of linear classifiers, the two hypotheses that satisfy (i.e., with $A_{D_u} = 0$) the above constraints are $h^\star$ and $-h_{\text{sp}}$, where $-h_{\text{sp}}(x) = 1 - \mathcal{I}\{x_2 < 0\} = \mathcal{I}\{x_2 > 0\}$. When considering the entire objective $\mathcal{L}_{D_t}^{\text{ERM}} + \alpha A_{D_u}$, only $h^\star$ minimizes the objective to 0, regardless of $\alpha$. Therefore, in this case, the D-BAT's solution corresponds to the ground truth function $h^\star$.

- When $r_{D_u} = 0.5$ (balanced), the unlabeled OOD data spans $\{[-1,1]\cup[0,1]\}\cup\{[-1,1]\cup[-1,0]\}$, i.e., all four quadrants. The diversification loss is, therefore:

$$
\begin{aligned}
A_{D_u} &= \mathbb{E}_{x\sim D_u}[-\log(P_{h_{\text{sp}}}(x;0)\cdot P_{h_2^{DB}}(x;1) + P_{h_{\text{sp}}}(x;1)\cdot P_{h_2^{DB}}(x;0))] \\
&= \mathbb{E}_{x\sim D_u}[-\log(1\cdot P_{h_2^{DB}}(x;1) + 0\cdot P_{h_2^{DB}}(x;0)) \mid x\in\{[0,1]\cup[0,1]\}]\cdot\frac{1}{4} \quad \text{(2nd quadrant)} \\
&+ \mathbb{E}_{x\sim D_u}[-\log(1\cdot P_{h_2^{DB}}(x;1) + 0\cdot P_{h_2^{DB}}(x;0)) \mid x\in\{[-1,0]\cup[0,1]\}]\cdot\frac{1}{4} \quad \text{(1st quadrant)} \\
&+ \mathbb{E}_{x\sim D_u}[-\log(0\cdot P_{h_2^{DB}}(x;1) + 1\cdot P_{h_2^{DB}}(x;0)) \mid x\in\{[-1,0]\cup[-1,0]\}]\cdot\frac{1}{4} \quad \text{(3rd quadrant)} \\
&+ \mathbb{E}_{x\sim D_u}[-\log(0\cdot P_{h_2^{DB}}(x;1) + 1\cdot P_{h_2^{DB}}(x;0)) \mid x\in\{[0,1]\cup[-1,0]\}]\cdot\frac{1}{4} \quad \text{(4th quadrant)}
\end{aligned}
$$
(5)

The hypothesis $h_2$ which minimizes Eq. 5 requires $P_{h_2^{DB}}(x;1) = 1$ for $x$ in the 1st & 2nd quadrants, and $P_{h_2^{DB}}(x;0) = 1$ for $x$ in the 3rd & 4th quadrants. The only hypothesis that satisfies these conditions is $-h_{\text{sp}}$. Although $-h_{\text{sp}}$ doesn't minimize $\mathcal{L}_{D_t}^{\text{ERM}}$, we note that, given that the hypothesis function is hard-margin, any data point $x$ in the 1st and 4th quadrants drives the diversification loss to positive infinity if $h_2^{DB}(x) \neq -h_{\text{sp}}(x)$. This is because of the $-\log$ in the diversification loss. Therefore, in this case, D-BAT's solution is $-h_{\text{sp}}$ regardless of $\mathcal{L}_{D_t}^{\text{ERM}}$ and $\alpha$. In practice (soft-margin regime), given that the $\alpha$ parameter is large enough to enforce diversification, this also holds as we empirically verified in Fig. 2-Left.

- The cases where $0 < r_{D_u} < 0.5$ can be straightforwardly extended from the above two cases. Indeed, as said above, any unlabeled data point in the 1st and 4th quadrants drives the diversification loss to be positive infinity if $h_2^{DB} \neq -h_{\text{sp}}$, and, thus, $h_2^{DB} = -h_{\text{sp}}$. Note, that this "phase transition" arises in theory as we consider all the points from $D_{\text{ood}}$ appearing in the 1st and 4th quadrants, i.e., $\{[R(r_{D_u}),0]\cup[0,1]\}$ and $\{[0,-R(r_{D_u})]\cup[-1,0]\}$. In practice, when $D_{\text{ood}}$ contains only some samples from these regions, the $h_2^{DB}$ will rotate counterclockwise as we increase $r_{D_u}$ from 0 to 0.5, starting at $h_{\text{GT}}$ and ending at $-h_{\text{sp}}$ as seen in the empirical experiment shown in Fig. 2.

Overall, when $r_{D_u} = 0$, there are $h^\star$ and $-h_{\text{sp}}$ minimizing diversification loss with minimum 0, and only $h^\star$ minimizes the whole loss (ERM + diversification loss). On the other hand, when $r_{D_u} = 0.5$, D-BAT finds $-h_{\text{sp}}$ that minimizes diversification loss but violates the ERM objective, regardless of the choice of $\alpha$.

**DivDis-Seq.** The DivDis diversification loss is

$$
A_{D_u}(h_{\text{sp}}, h_2^{DD}) = D_{\text{KL}}(P_{(h_{\text{sp}}, h_2^{DD})} || P_{h_{\text{sp}}} \otimes P_{h_2^{DD}}) + \lambda D_{\text{KL}}(P_{h_2^{DD}} || P_{D_t}).
$$

The first term on the right-hand side is the mutual information between $h_{\text{sp}}$ and $h_2^{DD}$. Minimizing mutual information on the unlabeled OOD data $D_u$ yields a hypothesis $h_2^{DD}$ that disagrees with $h_{\text{sp}}$ on $\frac{N_U}{2}$ data points while agreeing on the other $\frac{N_U}{2}$ (where $N_U$ is the size of unlabeled OOD data). In a finite sample size, this is equivalent to the hypotheses being statistically independent.

The second term is the KL-divergence between the class distribution of $h_2$ on unlabeled data $D_u$ and the class distribution of $D_t$. In this setting, an hypothesis $h_2$ minimizing this metric simply needs to classify half of the samples in the first class and the other half in the second class. Therefore:

- When $r_{D_u} = 0$ (inversely correlated), minimizing $\mathcal{L}_{D_t}^{\text{ERM}}(h_2^{DD}) + \alpha A_{D_u}(h_{\text{sp}}, h_2^{DD})$ finds $h_2^{DD} = h(x;\frac{\pi}{4})$. Indeed, it is the only linear classifier that satisfies (minimizes to 0) both objectives. It

classifies all data points from $D_t$ correctly and 'half' disagrees (i.e., statistically independent) on 2nd and 3rd quadrants with $h_{\text{sp}}$, and classifies half of the unlabeled samples in each class.

- When $r_{D_u} = 0.5$ (balanced), minimizing $\mathcal{L}_{D_t}^{\text{ERM}}(h_2^{DD}) + \alpha A_{D_u}(h_{\text{sp}}, h_2^{DD})$ finds $h_2^{DD} = h(x; \frac{\pi}{2}) = h^\star$ as the only hypothesis satisfying both losses similar to the previous case.

- In general, for $0 \leq r_{D_u} \leq 0.5$, the classification boundary of $h_2^{DD}$ rotates counterclockwise (starting at $h(x; \frac{\pi}{4})$ for $r_{D_u} = 0$) as the spurious ration increases i.e. $h_2^{DD} = h(x; \beta(r_{D_u}))$, where $\beta(r) : [0, 0.5] \rightarrow [\frac{\pi}{4}, \frac{\pi}{2}]$ is an increasing function of $r$. More precisely, $\beta(r) = \frac{\pi}{2} - \arctan(\frac{1-2r}{1-r})$. This is the solution that satisfies both losses, similar to the previous cases. The decision line lies in the 2nd and 3rd quadrants, and, therefore classifies the labeled training data correctly. The angle can be easily derived to satisfy the constraint that $h_2^{DD}$ and $h_{\text{sp}}$ agree only on half of the unlabeled OOD data. Since $\beta(r)$ is strictly increasing, it is only when $r_{D_u} = 0.5$ the solution of DivDis-Seq coincides with the ground truth $h^\star = h(x; \frac{\pi}{2}) = h(x; \beta(0.5))$.

In this setting, this means DivDis-Seq only finds $h^\star$ when the unlabeled OOD data is balanced.

$\square$

**Conclusion.** Overall, we see that the two methods find $h^\star$ in completely different conditions, which is consistent with the observation in Fig. 2 and thus calls for attention on one of the key components – the spurious ratio of unlabeled OOD data $r_{D_u}$.

**DivDis.** Since simultaneous training introduces a more complex interaction between the two hypotheses, we do not provide proof for DivDis. In Appendix C, we give empirical results on the 2D example showing that DivDis only finds $h^\star$ when $r_{D_u} = 0.5$, as DivDis-Seq does, but the solutions are different for other values of the spurious ratio.

## B    RESULTS FOR TRAINING MLPS ON 2D TASK

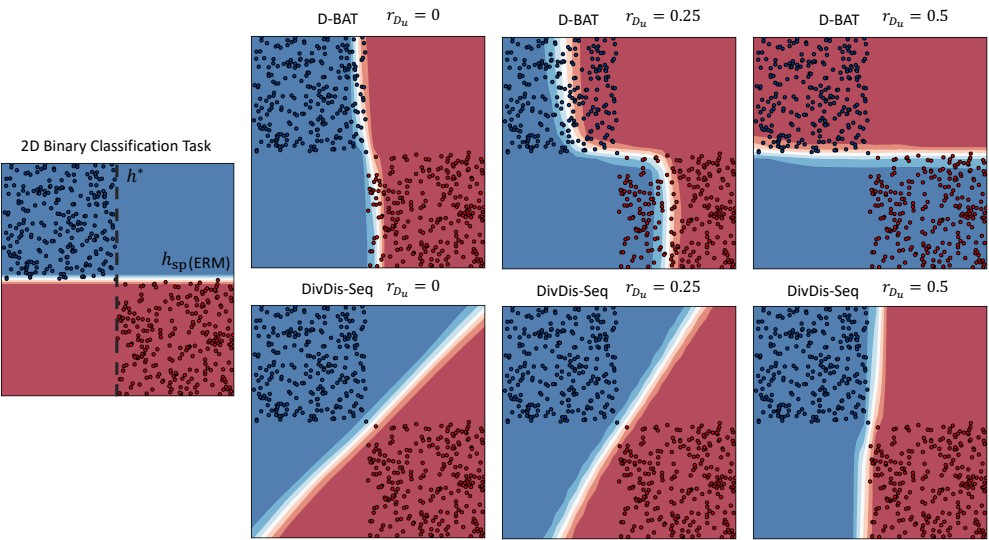

Figure 5: **Performance of diversification is highly dependent on unlabeled OOD data (2D example + MLP).** The unlabeled OOD data points are *not* shown in the plots. **Left:** the labeled training data $D_t$, the ground truth function $h^\star$ and the spurious function $h_{\text{sp}}$. **Right:** the second hypothesis for D-BAT and DivDis-Seq under different spurious ratios of unlabeled OOD data.

In this section, we investigate whether the influence of the spurious ratio of unlabeled OOD data shown in Proposition 1 still holds when the learning algorithm is more flexible. Specifically, we use the same 2D settings, described in Sec. 4 and Appendix A, but we train a multilayer perceptron (MLP) instead of the linear classifier. The MLP consists of 3 fully connected layers (with a width of 40) and has ReLU as the activation function.

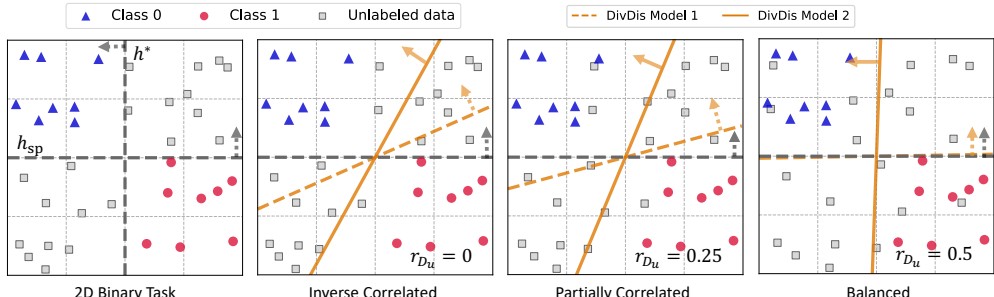

Figure 6: **(Complementary results for Fig. 2-Left) Learning Dynamics of DivDis (Lee et al., 2023) on 2D binary task. Left: Top-left quadrant**: The 2D binary classification task. **Other quadrants**: The two diverse hypotheses (arrows are normal vectors) found by DivDis with varied spurious ratios of unlabeled OOD data $r_{D_u} = \{0, 0.25, 0.5\}$ (from inversely correlated to balanced).

As shown in Fig. 5, we observe that D-BAT (Pagliardini et al., 2023) finds $h^\star$ only in inversely correlated unlabeled OOD data, while DivDis-Seq, on the contrary, finds $h^\star$ under balanced unlabeled OOD data, which is consistent with the Proposition 1. Indeed, for D-BAT, when $r_{D_u} = 0.25$ or $r_{D_u} = 0.5$, the diversification loss contradicts the cross-entropy loss on the labeled training data, causing misclassification of the training data. On the contrary, DivDis-Seq's boundary rotates *counterclockwise*, and its diversification loss causes no contradiction with the cross-entropy training loss.

## C  RESULTS FOR DIVDIS ON 2D BINARY TASK

In Sec. 4.1, we show how varying the spurious ratio influences the learning dynamics of DivDis-Seq and D-BAT. For completeness, we also provide results for DivDis (Fig. 6). The experimental setup is the same as in Sec. 4, with 0.5k / 5k training and unlabeled OOD data (with varied spurious ratio). Similarly, the hypothesis space is restricted to linear classifiers. Because DivDis optimizes simultaneously (i.e., there is no *first/second* model), we do not fix the first classifier to $h_{\mathrm{sp}}$ contrary to what was done for D-BAT and DivDis-Seq.

As shown in Fig. 6, DivDis does not find the true hypothesis $h^\star$ when the unlabeled OOD data is inversely correlated. On the contrary, it recovers $h_{\mathrm{sp}}$ and $h^\star$ when the unlabeled OOD data is balanced. Thus, DivDis and DivDis-Seq share similar learning dynamics when $r_{D_u} = 0.5$.

## D  MORE DETAILS, RESULTS AND DISCUSSION FOR SEC. 4.2

In Sec. 4.2, we demonstrate on *real-world image data* that one of the key factors influencing the performance of diversification methods is the distribution of the unlabeled OOD data (more specifically, the spurious ratio $r_{D_u}$). Here we provide more details for the experimental setup, and results (in Fig. 9) for M/F (Pagliardini et al., 2023) dataset (where MNIST and Fashion-MNIST (Xiao et al., 2017) are concatenated). We then provide the results on CelebA (Liu et al., 2015; Lee et al., 2023) as verification on a large-scale dataset, as shown in Tab. 4.

**Dataset & model details.** We investigate two datasets: M/C and M/F, where:

- In the training set (Fig. 7), the spurious dataset (MNIST) completely correlates with the "true" or semantic dataset (CIFAR-10 (Krizhevsky & Hinton, 2009) or Fashion-MNIST (Xiao et al., 2017)). Specifically, in M/C, MNIST 0s and 1s always concatenate with cars and trucks, respectively. In M/F, MNIST 0s and 1s always concatenate with coats and dresses, respectively.

- For the unlabeled data $D_u$, where D-BAT & DivDis(-Seq)'s hypotheses make diverse predictions, the spurious ratio $r_{D_u}$ changes, exposing that the performance of diversification is highly dependent on the unlabeled data.

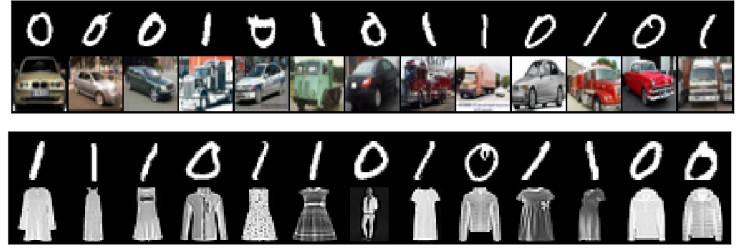

Figure 7: **Training data samples** from M/C (Top) and M/F (Bottom). Note that $D_t$ is completely spurious correlated ($r_{D_t} = 1$). Thus, MNIST 0s/1s match with CIFAR-10 cars/trucks or F-MNIST coats/dresses.

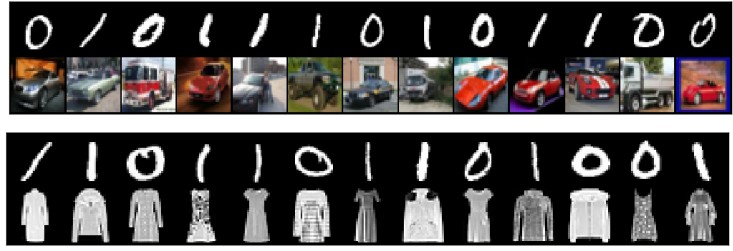

Figure 8: **Test data samples** from M/C (Top) and M/F (Bottom). Note that $D_{\text{ood}}$ is balanced ($r_{D_{\text{ood}}} = 0.5$).

- We can straightforwardly vary the $r_{D_u}$ by changing the rules of concatenation in $D_u$. Specifically, take M/C as an example, we first take the samples of 0s and 1s from MNIST, as well as cars and trucks from CIFAR-10, and we make sure they have the same size and are shuffled. Then, according to spurious ratio $r_{D_u}$, we randomly select a $r_{D_u}$ proportion of the samples from MNIST (0s / 1s) and CIFAR-10 (cars/trucks), and concatenate 0s with cars and 1s with trucks (so that the semantic feature is correlated with the spurious feature in $r_{D_u}$ of samples). We finally concatenate the remaining $1 - r_{D_u}$ proportion of samples oppositely (i.e., 0s with trucks and 1s with cars).
- $r_{D_u} = 0$ means *inversely correlated* (all images are 0s/trucks or 1s/cars).
- $r_{D_u} = 0.5$ means *balanced* (half of the 0s are concatenated with cars and the other half is concatenated with trucks, half of the 1s are concatenated with trucks and the other half is concatenated with cars).
- $r_{D_u} = 1$ means *completely spurious* (all images are 0s/cars or 1s/trucks).
- The test data is a hold-out balanced OOD data $D_{\text{ood}}$ (Fig. 8), i.e., $r_{D_{\text{ood}}} = 0.5$, in which there is no spurious correlation between the MNIST and target dataset (either CIFAR-10 or Fashion-MNIST), and the labels are assigned according to CIFAR-10 (in M/C) and Fashion-MNIST (in M/F).

We train a LeNet (Lecun et al., 1998), which contains 2 convolutional layers and 3 linear layers. Following Pagliardini et al. (2023) setup, depending on the dataset, we modify the number of channels and input / output sizes of the linear layers. We summarize these parameters in Tab. 3.

|     | Conv 1 | Conv 2 | Linear 1 | Linear 2 | Linear 3 | Pooling |
| --- | --- | --- | --- | --- | --- | --- |
| M/C | 3, 32, 5 | 32, 56, 5 | 2016, 512 | 512, 256 | 256, 2 | Average |
| M/F | 1, 6, 5 | 6, 16, 5 | 960, 120 | 120, 84 | 84, 2 | Max |

Table 3: **LeNet's parameters in Sec. 4.2**. For Conv layers, the numbers represent the input channel, output channel and kernel size. For linear layers, the numbers are input and output sizes, respectively.

**Results on M/F dataset.** In the same manner of Fig.2, we show results on M/F dataset in Fig. 9-Right. We see a similar trend as Fig.2:

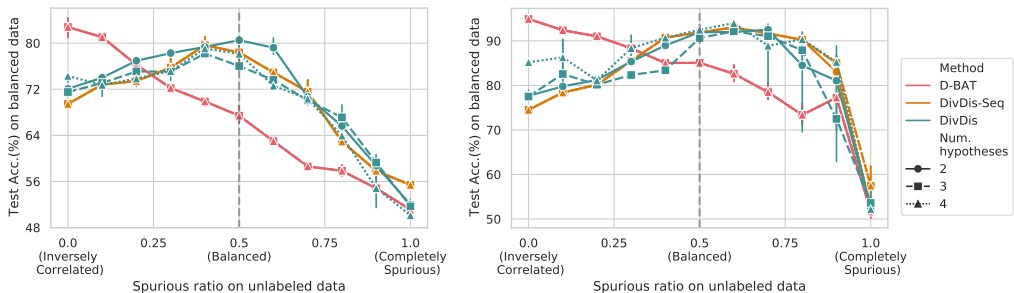

Figure 9: **Performance of diversification is highly dependent on unlabeled OOD data (M/C and M/F datasets).** The test accuracy is measured on balanced data $D_{\text{ood}}$ (i.e. $r_{D_{\text{ood}}} = 0.5$, no spurious correlation). **Left**: Test accuracy of D-BAT & DivDis(-Seq) on MNIST/CIFAR-10 for varied spurious ratios $r_{D_u}$. **Right**: Test accuracy of D-BAT & DivDis(-Seq) on MNIST/Fashion-MNIST for varied spurious ratios $r_{D_u}$.

- When $r_{D_u} \in [0, 0.5]$, (inversely correlated to balanced), the results match our observations made in .
- When $r_{D_u} \in [0.5, 1.0]$ (balanced to completely spurious), both on M/C and M/F, all methods have more and more difficulty to diversify and use the semantic features. Indeed, the unlabeled OOD data distribution $D_u$ gets increasingly closer to the training distribution $D_t$, thus we cannot expect OOD generalization.

Overall the synthetic 2D binary task section, M/C, and M/F experiments suggest that, in practice, across different datasets, diversification methods' behavior and solutions are highly dependent on the spurious ratio of unlabeled OOD data.

**Discussion on the $\alpha$ hyperparameter.** In the above experiments on both datasets, we use large coefficients $\alpha$ for diversification losses ($A_{D_u}$ in Eq. 2) as 5 / 50 / 50 for D-BAT / DivDis / DivDis-Seq, in order to study the behavior of these methods when the diversity objective is fully optimized.

In Fig. 10-Left, we further show results for different values of $\alpha$. We observe that tuning $\alpha$ is *not* sufficient to compensate for the misalignment between the unlabeled OOD data and the diversification loss, and the performance for both methods has the same trend. Specifically, larger $\alpha$ gives better test accuracy in general, as shown in Fig. 10-Left. In Fig. 10-Right, we select the best $\alpha$ for each scenario (i.e., each spurious ratio of unlabeled OOD data), and observe no meaningful difference in behavior (compared to Fig. 2 and Fig. 9). Therefore, a conclusion similar to Proposition 1 still holds: even when tuning $\alpha$ for each unlabeled OOD data setting (i.e. spurious ratio), D-BAT performs best when the unlabeled data is inversely correlated, while DivDis performs best when the unlabeled data is balanced. This suggests that a practitioner might not be able to compensate for a misalignment between unlabeled data and diversification loss by tuning the hyperparameter $\alpha$.

**Results on CelebA-CC dataset.** In Tab. 4, we further show results on a large-scale real-world dataset, namely CelebA-CC (Liu et al., 2015; Lee et al., 2023). CelebA-CC is a variant of CelebA, introduced by (Lee et al., 2023), where the training data semantic attribute is completely correlated with the spurious attribute. Here, gender is used as the spurious attribute and hair color as the target. We take D-BAT and DivDis-Seq (for fair comparison on sequential training), and show their test accuracy on different degrees of spurious ratio of unlabeled OOD data ($r_{D_u} = \{0.0, 0.5, 1.0\}$). Consistent with our previous observations, the results show that D-BAT performs the best when $r_{D_u} = 0.0$, and DivDis-Seq performs the best when $r_{D_u} = 0.5$.

## E  PROOF OF PROPOSITION 2

We first remind our proposition:

**Proposition 2.** *For $K = (2|D_{\text{ood}}| - 1)$ and $h^*$ the OOD labeling function, there exists a set of diverse $K$ hypotheses $h_1, ..., h_K$, i.e., $A_{D_{\text{ood}}}(h_i, h_j) = |\{x \in D_{\text{ood}} : h_i(x) = h_j(x)\}|/|D_{\text{ood}}| \leq 0.5 \quad \forall i, j \in \{1, ..., K\}, i \neq j$ and it holds that $\max_{h' \in h_1, ..., h_K} \text{Acc}(h^*, h') \leq 0.5$.*

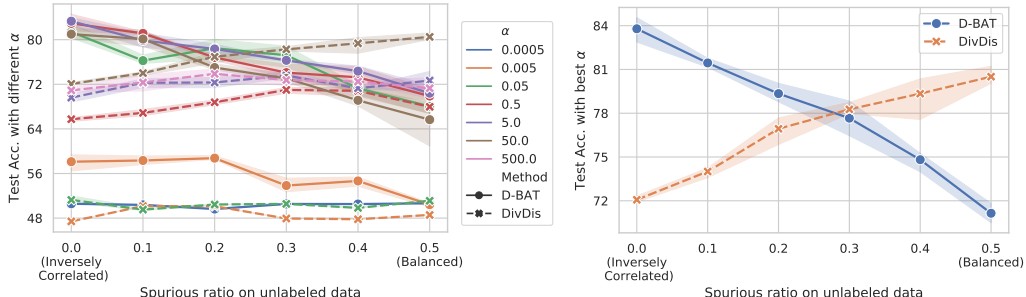

Figure 10: **Tuning $\alpha$ is _not_ sufficient to compensate for the change in performance when the spurious ratio of unlabeled OOD data changes.** The test accuracy is measured on balanced data $D_{\text{ood}}$ of MNIST/CIFAR-10. **Left**: Test accuracy of D-BAT and DivDis with varied spurious ratios of unlabeled OOD data, where various $\alpha$ are considered. **Right**: Test accuracy with the best $\alpha$ for D-BAT and DivDis, where for each spurious ratio, the best $\alpha$ is selected w.r.t the accuracy on a hold-out balanced validation data.

|  | $r_{D_u} = 0.0$ | $r_{D_u} = 0.5$ | $r_{D_u} = 1.0$ |
|---|---|---|---|
| D-BAT | $84.6_{\pm 0.3}$ | $82.8_{\pm 0.2}$ | $74.6_{\pm 0.6}$ |
| DivDis-Seq | $84.8_{\pm 0.2}$ | $86.1_{\pm 0.1}$ | $73.2_{\pm 0.4}$ |

Table 4: **Verification of the conclusion in Proposition 1 with CelebA-CC.** Gender is used as the spurious attribute and hair color as the target. We report average test accuracy. The trends of accuracy are consistent with what was shown in Fig.9, indicating that the conclusion that D-BAT reaches its optimal when $r_{D_u} = 0.0$ and DivDis(-Seq) reaches its optimal when $r_{D_u} = 0.5$ can **extend to a much larger and more realistic dataset**. DivDis-Seq is used here for a fair comparison.

This formulation covers our two methods of interest, D-BAT (Pagliardini et al., 2023) and DivDis (Lee et al., 2023). Indeed, the maximum agreement is upper-bounded by 0.5. For DivDis, the optimal solution has a maximum agreement of 0.5, as seen in Appendix A. For D-BAT, the optimal solution has the lowest agreement possible. Indeed, for $K = 2$, the optimal solution has $A_{D_{\text{ood}}}(h_1, h_2) = 0$. Thus, both methods optimal solutions are covered when upper-bounding the maximum agreement by 0.5 (as long as $K \leq (2|D_{\text{ood}}| - 1)$).

We prove the existence of a diverse set of $K$ hypotheses, satisfying the condition of Proposition 2, using a classic construction from coding theory, called the Hadamard code (Bose & Shrikhande, 1959).

**Terminology.** We first make explicit the equivalence between a hypothesis space and coding theory terminology. In binary classification, a labeling function or hypothesis $h_i$ on $D_{\text{ood}}$ is a binary codeword (vector) of fixed length $N$, where $N = |D_{\text{ood}}|$. A set of $K$ hypotheses is now referred to as a _code_ $C$ of size $K$. We define the Hamming distance between codewords $h_i, h_j$ as $d(h_i, h_j) = \sum_{k=1}^{N} \mathcal{I}[h_i(k) \neq h_j(k)]$ where $\mathcal{I}$ is the indicator function and $h_i(k)$ is the hypothesis prediction on the $k$th data point from $D_{\text{ood}}$. The Hamming distance between two equal-length codewords of symbols is the number of positions at which the corresponding symbols are different.

The agreement between two hypotheses $h_i, h_j$ can now be rewritten using the Hamming distance as $A_{D_{\text{ood}}}(h_i, h_j) = \frac{1}{N} \sum_{k=1}^{N} \mathcal{I}[h_i(k) = h_j(k)] = \frac{1}{N}(N - \sum_{k=1}^{N} \mathcal{I}[h_i(k) \neq h_j(k)]) = 1 - \frac{d(h_i, h_j)}{N}$. Similarly, the accuracy can also be rewritten as $\text{Acc}(h^*, h') = A_{D_{\text{ood}}}(h^*, h') = 1 - \frac{d(h^*, h')}{N}$.

_Proof._

We first use the fact that there exists a binary code $C$ with minimum distance $d^* = \min_{x,y \in C, x \neq y} d(x, y) = \frac{N}{2}$ and $|C| = 2N$. This binary code is the Hadamard code (Bose & Shrikhande, 1959; Rudra, 2007), also known as Walsh code. This binary code has $2N$ codewords of length $N$ and has the minimal distance of $\frac{N}{2}$.

We show now that we can modify the Hadamard code $C$ to obtain another code $C'$ with equivalent properties and $h^* \in C'$. For $C'$, it then holds that $\max_{h' \in C'} \mathrm{Acc}(h^*, h') \leq 0.5$, as it was shown above that $\mathrm{Acc}(h^*, h') = 1 - \frac{d(h^*, h')}{N} \leq 1 - \frac{d^*}{N} = 1 - \frac{N/2}{N} = 0.5$. Further, we show how to construct such $C'$.

Let $h_1$ be the first codeword of $C$. Let us now define a function (or transformation) $f(h) : \{0, 1\}^N \to \{0, 1\}^N$ such that $f(h_1) = h^*$, i.e $f$ transforms $h_1$ into $h^*$. Since we are dealing with binary vectors, the function $f(h)$ can be broken down into individual bit flips i.e.

$$
f(h)(i) = \begin{cases} h(i) & \text{if } h_1(i) = h^*(i) \\ 1 - h(i) & \text{if } h_1(i) \neq h^*(i) \end{cases}
$$

Applying $f$ to all codewords in code $C$ gives us a new code $C' = \{h \in C : f(h)\}$ and $h^* \in C'$. This operation maintains the minimum distance $d = \frac{N}{2}$ since:

$$
\begin{aligned}
d(f(c_i), f(c_j)) &= \sum_{k=1}^{N} \mathcal{I}[f(h_i)(k) \neq f(h_j)(k)] \\
&= \sum_{\{k: \, h_1(k) = h^*(k)\}} \mathcal{I}[h_i(k) \neq h_j(k)] \\
&\quad + \sum_{\{k: \, h_1(k) \neq h^*(k)\}} \mathcal{I}[1 - h_i(k) \neq 1 - h_j(k))] \quad \text{(by definition of } f) \\
&= \sum_{\{k: \, h_1(k) = h^*(k)\}} \mathcal{I}[h_i(k) \neq h_j(k)] \\
&\quad + \sum_{\{k: \, h_1(k) \neq h^*(k)\}} \mathcal{I}[h_i(k) \neq h_j(k))] \\
&= \sum_{k=1}^{N} \mathcal{I}[h_i(k) \neq h_j(k)] \\
&= d(h_i, h_j)
\end{aligned}
$$

Therefore, $C'$ is the code satisfying all the conditions of Proposition 2.

As was shown before, a binary code $C$ is equivalent to a set of $|C|$ hypotheses with the same properties. Thus, with $N = |D_{\mathrm{ood}}|$, the above construction gives us a set of $(2|D_{\mathrm{ood}}| - 1)$ ($2N$ minus the true labeling $h^*$) hypotheses satisfying the constraints of Proposition 2. This concludes our proof.

$\square$

## Extension to multi-class classification

We used the mathematical framework of coding theory and a classical result from it, the Hadamard code (Bose & Shrikhande, 1959), to prove Proposition 2, specifically for binary hypotheses. However, in coding theory, it has not been proven yet whether codes with similar "nice" properties, similar to Hadamard's, exist for any $q$-ary codes i.e. for hypotheses with $q$ possible classes. One exception is when $q$ is a prime number.

**Proposition 3.** *Let $q$ be the number of classes and a prime number. Let $m \in \mathbb{N}^+$ s.t. $|D_{\mathrm{ood}}| = q^m$. Then, for $K = (q \cdot |D_{\mathrm{ood}}| - 1)$ and $h^*$ the OOD labeling function, there exists a set of diverse $K$ $q$-ary hypotheses $h_1, ..., h_K$, s.t., $A_{D_{\mathrm{ood}}}(h_i, h_j) = |x \in D_{\mathrm{ood}} : h_i(x) = h_j(x)|/|D_{\mathrm{ood}}| \leq \frac{1}{q} \quad \forall i, j \in 1, ..., K, i \neq j$, and it holds that $\max_{h' \in h_1, ..., h_K} \mathrm{Acc}(h^*, h') \leq \frac{1}{q}$*

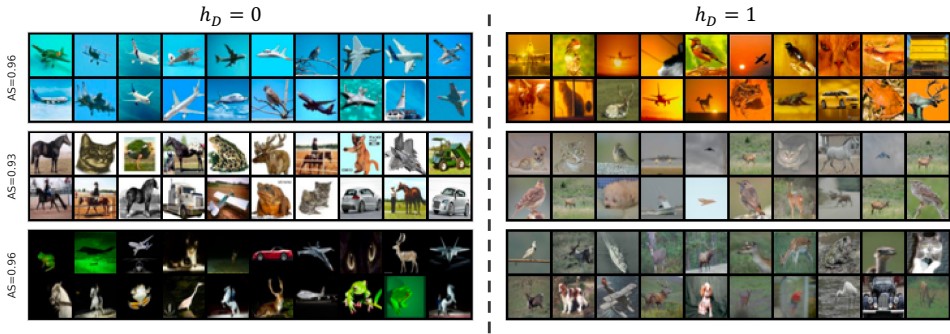

Figure 11: **Examples of high-AS hypotheses discovered in Atanov et al. (2022).** Each hypothesis is illustrated by exemplar images from each class as labeled be the corresponding discovered hypothesis $h_D$. Neural networks can generalize by training on the $h_D$'s labeling.

*Proof.* Using a similar argument from the proof of Proposition 2, (Stepanov, 2006; 2017) tells us that for any $|D_{\text{ood}}| = q^m$ where $m \in \mathbb{N}^+$, we can find a code similar to Hadamard's with minimum distance equal to $\frac{N(q-1)}{q}$ and cardinality equal to $q^{m+1} = q \cdot |D_{\text{ood}}|$. By removing the semantic hypothesis from the count, we obtain that Proposition 3 holds for $K = (q \cdot |D_{\text{ood}}| - 1)$. □

## F AGREEMENT SCORE AND IMPLICIT BIAS OF DIVERSE HYPOTHESES

### F.1 AGREEMENT SCORE AND TASK DISCOVERY (ATANOV ET AL., 2022)

In this section, we introduce more details on the background of Atanov et al. (2022), as well as how we leverage the findings from it.

**Agreement score as a measure of inductive bias alignment.** We use the agreement score (AS) (Atanov et al., 2022; Hacohen et al., 2020; Jiang et al., 2022) to measure the alignment between the found hypotheses and the inductive biases of a learning algorithm. It is measured in the following way: given a training dataset $D_t$ labeled with a true hypothesis $h^*$, unseen unlabeled data $D_{\text{ood}}$, and a neural network learning algorithm $\mathcal{A}$, train two networks from different initializations on the same training data, resulting in two hypotheses $h_1, h_2 \sim \mathcal{A}(D_t, h^*)$, and measure the agreement between these two hypotheses on $D_{\text{ood}}$:

$$\text{AS}_{\mathcal{A}}(h^*; D_t, D_{\text{ood}}) = \mathbb{E}_{h_1, h_2 \sim \mathcal{A}(D_t, h^*)} \mathbb{E}_{x \sim D_{\text{ood}}} [h_1(x) = h_2(x)] \qquad (6)$$

Recent works (Atanov et al., 2022; Baek et al., 2022) show that the AS correlates well with how well a learning algorithm $\mathcal{A}$ generalizes on a given training task represented by a hypothesis $h$. Indeed, high AS is a necessary condition for generalization (Atanov et al., 2022) (different outcomes of $\mathcal{A}$ have to at least converge to a similar solution). Finally, a learning algorithm will generalize on a labeling if the labeling is aligned with the learning algorithm's inductive biases, thus, we use AS as a measure of how well a given hypothesis $h$ is aligned with the inductive biases of $\mathcal{A}$.

**Task Discovery.** Atanov et al. (2022) use bi-level optimization (also called meta-optimization) to optimize the agreement score (i.e., Eq. 6) and discover, on any dataset, high-AS hypotheses (tasks in the terminology of Task Discovery) that a given learning algorithm can generalize well on. They show that there are many *diverse* high-AS hypotheses different from semantic human annotations. In Fig. 11, we show examples of the high-AS hypotheses discovered for the ResNet18 architecture on CIFAR-10 (Krizhevsky & Hinton, 2009).

**Adversarial dataset splits (Fig. 12).** Atanov et al. (2022) also introduces the concept of adversarial dataset splits, which is a train-test dataset partitioning such that neural networks trained on the training set fail to generalize on the test set. To do that, they induce a spurious correlation between a high-AS discovered hypothesis $h_D$ and the (target) semantic hypothesis $h^\star$ on the training data, and the opposite correlation on the test set. Specifically, they select data points as training set $D_t$, such that a discovered high-AS hypothesis $h_D$ (specifically, $\text{AS}(h_D) > \text{AS}(h^\star)$) completely spurious correlates with $h^\star$, i.e. $\{x \in D_t : h_D(x) = h^\star(x)\}$. The test set $D_{\text{test}}$ is constructed such that the

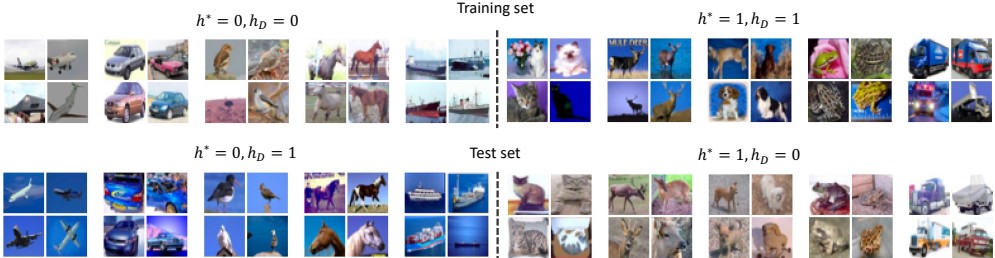

Figure 12: **Illustration of an adversarial split introduced in (Atanov et al., 2022).** A high-AS discovered hypothesis $h_D$ 'spuriously correlated' with semantic hypothesis $h^\star$ on the training set, but inversely correlated with $h^\star$ on test set. Training ERM on the training set and evaluating on test set give $< 20\%$ test accuracy (according to (Atanov et al., 2022)'s Fig. 7). In the context of this work, the test set in a given adversarial split is inversely correlated ($r = 0$).

two hypotheses are inversely correlated, i.e., $\{x \in D_{test} : h_D(x) \neq h^\star(x)\}$. Theoretically, a NN trained on such a training set $D_t$ should learn the hypotheses with a higher AS, i.e., $h_D$, which would lead to a low accuracy when tested on $D_{\text{test}}$. This was indeed shown to hold in practice, where the test accuracy drops from $0.8$ for a random split to $0.2$ for an adversarial split.

Adversarial splits, therefore, show that neural networks favor learning the task with a higher AS (the background color in the case of Fig. 12) when there are two hypotheses that can 'explain' the training data equally well. In this work, we refer to this 'preference' as an alignment between the neural network and $h_D$. This creates a controllable testbed for studying the effect of spurious correlations on NN training, which we also adopt in our study.

### F.2 DIVERSIFICATION FINDS HYPOTHESES ALIGNED WITH INDUCTIVE BIASES

*Disclaimer: For an introduction on agreement score, we refer to Appendix F.1*

In this section, we study how the diversification process is biased in practice by the inductive biases of the chosen learning algorithm. Specifically, using agreement score, we demonstrate that D-BAT and DivDis find hypotheses that are *not only diverse but aligned with the inductive bias of the learning algorithm*.

**Experimental setup**

**CIFAR-10.** we build on top of the adversarial splits and construct a CIFAR-10 data split with complete spurious correlation on the training data and a balanced (no spurious correlation) unlabeled OOD data, as shown in Fig.13. This is a typical setting on which D-BAT (Pagliardini et al., 2023) and DivDis (Lee et al., 2023) apply. More precisely, $h^\star$ is a semantic binary classification on CIFAR-10, defined by choosing a 5 vs 5 split of the original 10 classes. We define the spuriously correlated CIFAR-10 data by using an arbitrary high AS binary labeling $h_D$ as the spurious hypothesis, similarly to "adversarial splits" introduced by (Atanov et al., 2022). There are two reasons for using this setting: one to easily control the data setup, and one for using (Atanov et al., 2022) as a reference of the AS for different hypotheses on CIFAR-10.

**Measuring the AS of found hypotheses.** For a given dataset with training data $D_t$, unlabeled OOD data $D_{\text{ood}}^U$, and test OOD data $D_{\text{ood}}$, we train a diversification method (without pretraining) to find multiple diverse hypotheses $h_1, \ldots, h_K$ and measure their agreement scores. More precisely, for each hypothesis $h_i$, we measure $\text{AS}_\mathcal{A}(h_i; D_t \cup D_{\text{ood}}^U, D_{\text{ood}})$ where $\mathcal{A}$ is the same learning algorithm (e.g. ResNet18) used to find the diverse hypotheses. This AS allows us to assess whether $h_i$ is labeling $D_{\text{ood}}$ randomly or in a way that aligns well with the inductive biases of the learning algorithms. We provide more details on the setting in Appendix F.2 and illustrate the dataset creation in Fig. 13.

**Implicit bias of diverse hypotheses.** Tab. 5 shows the agreement score of random hypotheses $h_{\text{R}}$ (true labels on $D_t$ but random labels on $D_{\text{ood}}$) and the diverse hypotheses found by both diversification methods. We observe a clear gap between the two, indicating that all diverse hypotheses label $D_{\text{ood}}$ in a structured non-random manner.

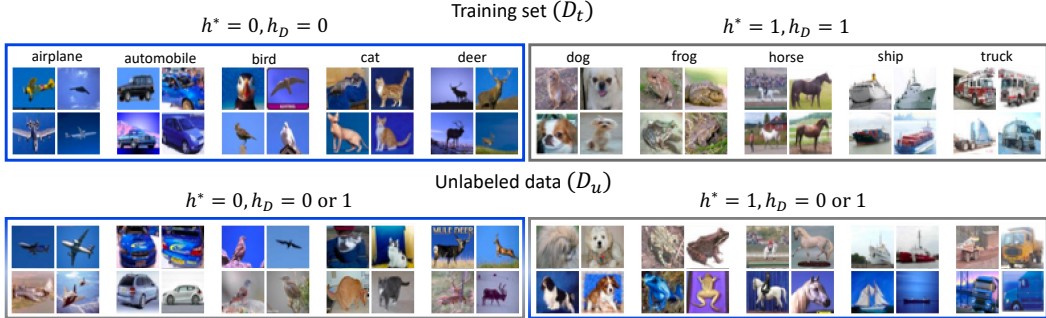

Figure 13: **Illustration of the constructed CIFAR-10 data with spurious correlation.** The semantic binary labeling $h^\star$ is defined by a 5 vs. 5 split of the original CIFAR-10 classes. Spurious hypothesis/feature $h_D$ (color) is predictive in the training set $D_t$, and non-predictive on unlabeled OOD data $D_u$.

| | CIFAR-10 | | Waterbirds | |
|---|---|---|---|---|
| Hypothesis | AS | Test Acc.(%) | AS | WG Acc.(%) |
| Semantic ($h^\star$) | 0.83 | 100.0 | 0.84 | 100.0 |
| Random ($h_R$) | $0.63_{\pm 0.01}$ | 50.0 | $0.55_{\pm 0.02}$ | 50.0 |
| D-BAT | $0.82_{\pm 0.01}$ | $60.1_{\pm 0.2}$ | $0.89_{\pm 0.01}$ | $25.3_{\pm 3.0}$ |
| DivDis | $0.81_{\pm 0.02}$ | $57.3_{\pm 0.5}$ | $0.91_{\pm 0.04}$ | $28.9_{\pm 4.8}$ |

Table 5: **The diverse hypotheses found by diversification methods have high AS** (i.e., aligned with the biases of the learning algorithm). Nonetheless, without additional inductive biases (e.g., the correct pretraining strategy, which is examined in Sec.5.2 and Tab. 7), they do not generalize. The test accuracy of CIFAR-10 is measured on hold-out balanced data ($D_{\text{ood}}$), and WG Acc. stands for Waterbirds worst-group test accuracy. Results are averaged over 3 seeds.

| | MLP | | ViT | |
|---|---|---|---|---|
| Hypothesis | AS | Test Acc.(%) | AS | Test Acc.(%) |
| Semantic ($h^\star$) | 0.80 | 100.0 | 0.76 | 100.0 |
| D-BAT | $0.89_{\pm 0.02}$ | $56.1_{\pm 0.4}$ | $0.90_{\pm 0.01}$ | $59.2_{\pm 0.3}$ |
| DivDis | $0.85_{\pm 0.02}$ | $58.3_{\pm 0.1}$ | $0.87_{\pm 0.02}$ | $57.7_{\pm 0.1}$ |

Table 6: **On CIFAR-10, D-BAT and DivDis also find high-AS hypotheses with MLP and ViT (Dosovitskiy et al., 2021).** Results are averaged over 3 seeds.

Measuring the agreement score of the true or semantic hypothesis $h^\star$ gives us an estimate of the expected AS value of a hypothesis aligned with the inductive bias of $\mathcal{A}$ (otherwise, we wouldn't expect $\mathcal{A}$ to be able to learn $h^\star$). We observe that the hypotheses found by DivDis and D-BAT have agreement scores similar to that of $h^\star$, indicating good alignment with the inductive biases of $\mathcal{A}$. Thus, optimizing Eq. 2, using neural networks as the learning algorithm, leads to diverse hypotheses *implicitly* biased towards those favored by its inductive biases. According to the definition of AS, such alignment is expected from a hypothesis found through empirical risk minimization (ERM), however, it is not expected from diverse hypotheses (as defined in Eq. 2), given that the additional diversification loss could destroy this alignment. This analysis sheds light on the process by which diverse hypotheses are found, and puts an emphasis on the choice of a good learning algorithm, which is crucial, as we show in the subsequent Sec. 5.2.

Similar to what is shown with ResNet in Tab. 5, in Tab. 6, we repeat the experiment with two different architectures, MLP and ViT (Dosovitskiy et al., 2021), on CIFAR-10. The diverse hypotheses found by D-BAT and DivDis have high AS. This demonstrates that our above conclusions also hold with different architectures.

| Hypothesis | AS | WG Acc.(%) |
|---|---|---|
| Semantic ($h^\star$) | 0.84 | 100.0 |
| Random ($h_R$) | $0.55_{\pm 0.02}$ | 50.0 |
| D-BAT (NP) | $0.89_{\pm 0.01}$ | $25.3_{\pm 3.0}$ |
| D-BAT (P) | $0.86_{\pm 0.02}$ | $59.1_{\pm 1.6}$ |
| DivDis (NP) | $0.91_{\pm 0.04}$ | $28.9_{\pm 4.8}$ |
| DivDis (P) | $0.86_{\pm 0.02}$ | $81.3_{\pm 2.2}$ |

Table 7: **D-BAT and DivDis find high-AS hypotheses on Waterbirds (Sagawa et al., 2020)**. The Agreement Score (AS) is measured on different hypotheses on Waterbirds. For D-BAT and DivDis, $K = 2$ hypotheses are considered. 'WG Acc.' stands for worst-group accuracy. "NP" signifies non-pretrained, "P" signifies ImageNet pretrained. Two ResNet-50 models are trained from scratch when measuring the AS. The best model performance is shown for DivDis. For D-BAT, we always show the performance of the second model. Results are averaged over 3 seeds.

**Diverse hypotheses cannot generalize without the correct pretraining**

As seen above, D-BAT and DivDis produce diverse hypotheses *implicitly* biased towards those favored by the inductive biases of its learning algorithm. Nonetheless, this implicit bias may not lead to OOD generalization, i.e., $h^* \notin \mathcal{H}_K^*$, as the test accuracies in Tab. 5 are found to be near the chance level.

In Tab. 7, on Waterbirds, we repeat the same experiment as in Tab. 5, with an additional variable. The ResNet50 model is either trained from scratch or starting from ImageNet-1k supervised pretraining weights. We can see that pretraining does not affect whether DivDis and D-BAT find high AS hypotheses, however it greatly influences the generalization capability of the found hypotheses. These results corroborate with Sec. 5.2 that the correct choice of inductive bias is crucial to unlock OOD generalization.

# G  RESULTS AND IMPLEMENTATION DETAILS OF SEC. 5.2

## G.1  EXPERIMENTAL DETAILS

**Remarks on D-BAT and DivDis.**

- All experiments were run using DivDis and D-BAT respective codebases to ensure closest reproducibility to their presented methods and results.
- DivDis' default setting is to augment the data while training. This option was disabled to ensure a fair comparison to D-BAT.
- For their results, DivDis rebuilt the Waterbirds(Sagawa et al., 2020) dataset from scratch. On the contrary, D-BAT used the one provided by the WILDS(Koh et al., 2021) library. To ensure a fair comparison, both methods were run using the latter version of the dataset.
- If not precised, all train, validation and test splits are taken as provided from Pagliardini et al. (2023); Lee et al. (2023) or WILDS.
- The best models are selected according to validation accuracy.

**Computational resources.** Each experiment can be run on a single A100 40GB GPU.

**Models.** If not precised, the model used in most experiments is ResNet50 (He et al., 2015). Otherwise, when using a Vision Transformer (ViT), we use a ViT-B/16[4] (Dosovitskiy et al., 2021). The last exception is for DivDis Camelyon17 (DenseNet121 (Huang et al., 2017)).

**DivDis parameters.** For Waterbirds variants, the optimizer is SGD, the number of epochs is 100, the learning rate is 0.001, the weight decay is 0.0001. The $\alpha$ parameter (referred as $\lambda$ in DivDis) was tuned over $\{0.1, 1, 10\}$. For Office-Home, the optimizer is SGD, the number of epochs is 50, the learning rate is 0.001, the weight decay is 0.0001, and the $\alpha$ parameter was tuned over $\{0.1, 1, 10\}$. For Camelyon17, the original best-performing setting from DivDis was used.

---

[4]https://pytorch.org/vision/main/models/generated/torchvision.models.vit_b_16.html

| | | From scratch | | Self-supervised | | | | | | Supervised | |
| --- | --- | --- | --- | --- | --- | --- | --- | --- | --- | --- | --- |
| | | Resnet50 | ViT-B/16 | SwAV | SIMCLRv2 | MoCo-v2 | ViT-MAE | ViT-Dino | Adv. robustness | Resnet50 IN | ViT-B/16 IN |
| OfficeHome | ERM (D-BAT $h_1$) | $10.4_{\pm0.9}$ | $5.1_{\pm5.4}$ | $20.6_{\pm1.4}$ | $43.9_{\pm0.6}$ | $25.2_{\pm1.5}$ | $\underline{61.2}_{\pm0.9}$ | $55.5_{\pm2.9}$ | $53.6_{\pm0.7}$ | $58.3_{\pm0.3}$ | $\mathbf{75.9}_{\pm0.6}$ |
| | D-BAT $h_2$ | $9.5_{\pm0.5}$ | $8.1_{\pm5.9}$ | $21.3_{\pm0.9}$ | $46.1_{\pm0.5}$ | $27.1_{\pm1.2}$ | $61.9_{\pm0.7}$ | $61.6_{\pm1.6}$ | $55.5_{\pm0.7}$ | $58.2_{\pm0.9}$ | $\mathbf{79.2}_{\pm0.4}$ |
| | DivDis (best) | $7.0_{\pm0.1}$ | $9.4_{\pm0.5}$ | $23.1_{\pm1.2}$ | $39.6_{\pm1.2}$ | $28.3_{\pm0.8}$ | $47.9_{\pm0.7}$ | $10.3_{\pm1.9}$ | $51.4_{\pm0.8}$ | $\underline{55.9}_{\pm0.6}$ | $70.7_{\pm1.6}$ |
| Waterbirds-CC | ERM (D-BAT $h_1$) | $8.6_{\pm2.1}$ | $13.5_{\pm6.7}$ | $7.0_{\pm0.3}$ | $9.2_{\pm3.3}$ | $35.6_{\pm3.8}$ | $11.9_{\pm0.6}$ | $9.2_{\pm0.8}$ | $20.9_{\pm5.1}$ | $\underline{30.1}_{\pm2.3}$ | $33.1_{\pm2.2}$ |
| | D-BAT $h_2$ | $15.0_{\pm5.1}$ | $8.6_{\pm6.8}$ | $22.6_{\pm6.6}$ | $15.4_{\pm6.6}$ | $36.1_{\pm5.9}$ | $55.5_{\pm2.8}$ | $47.6_{\pm5.0}$ | $49.7_{\pm9.8}$ | $\mathbf{67.7}_{\pm3.2}$ | $\underline{57.2}_{\pm3.7}$ |
| | DivDis (best) | $25.0_{\pm2.7}$ | $7.8_{\pm3.9}$ | $20.4_{\pm16.9}$ | $22.5_{\pm4.0}$ | $\underline{49.4}_{\pm10.3}$ | $38.9_{\pm7.3}$ | $11.2_{\pm4.7}$ | $47.1_{\pm0.8}$ | $\mathbf{70.5}_{\pm4.7}$ | $22.7_{\pm2.3}$ |

Table 8: **The performance of diversification methods is highly sensitive to the choice of architecture and pretraining method..** Average accuracy on Office-Home and worst-group accuracy on Waterbirds-CC for different pretraining methods. All methods are pretrained on ImageNet-1k. If not specified, the used model is ResNet50. Results are averaged over 3 seeds. **Bold** and underline stand for the best and second best, respectively.

| Dataset | Method | $K = 2$ | $K = 3$ | $K = 4$ | $K = 5$ |
| --- | --- | --- | --- | --- | --- |
| Waterbirds-CC | D-BAT (ViT-B/16 IN) | $57.1_{\pm3.7}$ | $45.0_{\pm1.4}$ | $45.5_{\pm1.7}$ | $44.5_{\pm1.8}$ |
| | DivDis (MoCO-v2) | $49.4_{\pm10.3}$ | $51.7_{\pm6.0}$ | $49.6_{\pm8.3}$ | $48.4_{\pm0.9}$ |
| Office-Home | D-BAT (ViT-MAE) | $61.9_{\pm0.7}$ | $62.6_{\pm0.1}$ | $60.8_{\pm0.4}$ | $61.7_{\pm0.7}$ |
| | DivDis (Resnet50 IN) | $55.9_{\pm0.6}$ | $54.6_{\pm0.1}$ | $53.6_{\pm0.4}$ | $53.1_{\pm0.2}$ |

Table 9: **Increasing the number of hypotheses, while using the second-best inductive bias, does not bridge the performance gap with the best inductive bias**. Best hypothesis performance for DivDis and the corresponding hypothesis performance for D-BAT are reported. The second-best inductive bias was chosen according to Fig. 3. Results are averaged over 3 seeds.

**D-BAT parameters.** For Waterbirds variants and Office-Home, the optimizer is SGD, the learning rate is 0.001, the weight decay is 0.0001. Given that D-BAT optimizes sequentially, the number of epochs is an important parameter to tune. We tuned over epochs $\in \{30, 100\}$ and $\alpha \in \{0.0001, 0.1\}$. For Camelyon17, the original D-BAT best-performing setting was used.

## G.2 COMPLETE RESULTS OF SEC. 5.2

We provide the full results for Fig. 3 (ERM baseline included) in Tab. 8. We also provide the accuracy of each new head of D-BAT for Tab. 1 in Tab. 9. Finally, in Fig. 14, we further show that DivDis does not scale well to larger K (e.g. $K = 64$) "out-of-the-box", and the performance drops as the number of hypotheses increases. Note that testing D-BAT in this regime would be prohibitively expensive.

## G.3 PRETRAINING STRATEGY AND ARCHITECTURE DETAILS.

In Fig. 3, we vary the pretraining method and architecture, and measure the effects on performance. We provide additional details here. If not precised, the methods use a ResNet-50(He et al., 2015) model. All 8 variations are pretrained on the ImageNet-1k(Russakovsky et al., 2015) dataset:

- Self-supervised
  - SwAV (Caron et al., 2020)
  - SimCLRv2 (Chen et al., 2020a)
  - MoCo-v2 (Chen et al., 2020b)
  - ViT-B/16 MAE (He et al., 2021)
  - ViT-B/16 Dino (Caron et al., 2021)
- Supervised
  - Adversarially robust classifiers (Salman et al., 2020).
  - Resnet50 IN (He et al., 2015), supervised pretraining on ImageNet-1k. This is the pretraining method used by (Lee et al., 2023; Pagliardini et al., 2023) in their papers.
  - ViT-B/16 IN (Dosovitskiy et al., 2021), supervised pretraining on ImageNet-1k.

**Experimental details** For the adversarially robust classifier, the L2-Robust ImageNet ResNet-50 ($\epsilon = 0.05$) model was chosen, following the advice of (Salman et al., 2020), as it is hypothesized that smaller values of $\epsilon$ tend work better on datasets where leveraging finer-grained features are necessary (i.e., where there is less norm-separation between classes in the input space), such as

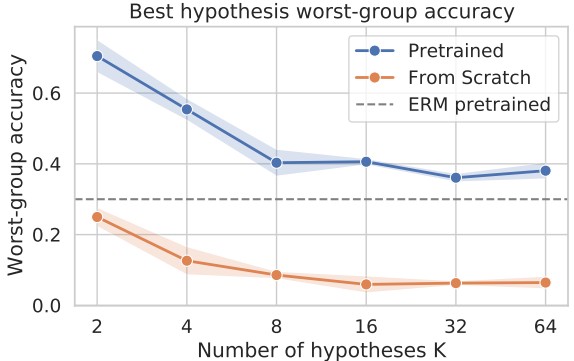

Figure 14: **Naively increasing the number of hypotheses in DivDis is detrimental to performance.** $y$-axis shows the worst-group accuracy of the best hypothesis on Waterbirds-CC. We use Resnet50 as a backbone. Pretraining is supervised pretraining on ImageNet-1k. "ERM pretrained" baseline stands for standard empirical risk minimization (ERM) without the usage of diversification. Results are averaged over 3 seeds. Preliminary investigation of this behavior suggests that the diversification loss is not optimized well when K grows large (e.g. $K > 8$) due to the averaging over all pairs of hypotheses, and that another aggregation function, e.g., maximum, could work better.

Waterbirds-CC or Office-Home. Each variation hyperparameters were tuned following the same procedure as described in G.1.

# H  DETAILED EXPERIMENTAL SETUP AND ADDITIONAL RESULTS FOR SEC. 5.3

In Sec.5.3, we demonstrate that using different inductive biases can drastically and predictably influence a diversification method. Here we provide more details on how we construct such examples. Additionally, in Tab. 11 we also provide results where the examples are constructed using a ViT-ResNet pair (instead of MLP-ResNet pair). Finally, we provide an extension Tab. 2 by showing how an inductive bias gradually gets favorable and vice versa through the transition of spurious ratios, in Fig. 15.

**Construction**

- **Prerequisite**: We consider a semantic (5-vs-5) binary classification task $h^\star$ on CIFAR-10 (Krizhevsky & Hinton, 2009) (i.e., airplane, automobile, bird, cat, deer original classes as class 1 and dog, frog, horse, ship, truck original classes as class 0).

- **Step 1 (Selecting hypotheses aligned with learning algorithms from (Atanov et al., 2022))**: We take two high-AS hypotheses (see Fig. 11 for examples of such hypotheses) discovered in (Atanov et al., 2022) for MLP and ResNet18 (He et al., 2015), where the hypotheses ($h_{\text{MLP}}$ and $h_{\text{RN}}$) satisfy Eq. 3:

$$\text{AS}_{\text{MLP}}(h_{\text{MLP}}) > \text{AS}_{\text{MLP}}(h_{\text{RN}}), \quad \text{AS}_{\text{RN}}(h_{\text{RN}}) > \text{AS}_{\text{RN}}(h_{\text{MLP}}),$$

  For example, this means the MLP hypothesis $h_{\text{MLP}}$ has a high AS when training MLPs but lower AS when training ResNet18s. Also, we ensure these two hypotheses have higher AS than the true hypothesis $h^\star$ to make sure that they are able to act as a spurious hypothesis.

- **Step 2 (Constructing training data where $h^\star$, $h_{\text{MLP}}$ and $h_{\text{RN}}$ completely correlates)**: As presented in Tab. 10, in $D_t$ row, we select the data points as training set $D_t$ such that $h^\star$, $h_{\text{MLP}}$ and $h_{\text{RN}}$ agree. As shown in (Atanov et al., 2022) by adversarial splits, when two hypotheses correlate with each other (i.e., their labels are the same on training data), a neural network tends to converge to the hypothesis with higher AS. Thus, combining with the conditions in step 1 (i.e., Eq. 3), training MLP on $D_t$ with ERM should converge to $h_{\text{MLP}}$ and training ResNet on $D_t$ with ERM to $h_{\text{RN}}$, which is illustrated in Tab. 2-Right.

|       | $h^\star$ | $h_{\text{MLP}}$ | $h_{\text{RN}}$ |       | $h^\star$ | $h_{\text{MLP}}$ | $h_{\text{RN}}$ |
|-------|-----------|------------------|-----------------|-------|-----------|------------------|-----------------|
| $D_t$ | 0 | 0 | 0 | $D_t$ | 0 | 0 | 0 |
|       | 1 | 1 | 1 |       | 1 | 1 | 1 |
| $D_u$ | 0 | 1 | 0 | $D_u$ | 0 | 0 | 1 |
|       | 0 | 1 | 1 |       | 0 | 1 | 1 |
|       | 1 | 0 | 0 |       | 1 | 0 | 0 |
|       | 1 | 0 | 1 |       | 1 | 1 | 0 |

Table 10: **The rules for constructing the data in Sec. 5.3**, covering the two cases in Tab. 2. **Left**: $h^\star \perp h_{\text{MLP}}$ on $D_u$. **Right**: $h^\star \perp h_{\text{RN}}$ on $D_u$. '$\perp$' means inversely correlated on the unlabeled OOD data $D_u$.

- **Step 3 (Further improving the alignment between the two hypotheses and their corresponding architectural inductive biases)**: This step is not necessary in general, but it allows us to find hypotheses that are better aligned with the inductive biases of the network. This is because the Task Discovery framework from (Atanov et al., 2022) might not provide globally optimal hypotheses. The improvement goes as follows: we update $h_{\text{MLP}}$ and $h_{\text{RN}}$ to further increase their AS for a better alignment with the corresponding learning algorithm (i.e., MLP and $h_{\text{MLP}}$, ResNet18 and $h_{\text{RN}}$). Specifically, we train with ERM an MLP and ResNet18 on $D_t$ and make predictions on all CIFAR-10 data except for the training data i.e. $D \setminus D_t$. We replace the old labels of $h_{\text{MLP}}$ and $h_{\text{RN}}$ on $D \setminus D_t$ by the new labels predicted by MLP and ResNet18. This step gives us higher AS hypotheses (thus more preferred by the given architecture) that satisfy Eq. 3 (equation also shown in Step 1).

- **Step 4 (Constructing unlabeled OOD data such that ResNet18 or MLP fails)**: As shown in Tab. 10, in $D_u$ row, we select data points with specific hypothesis labels as the unlabeled OOD data. By design, in Tab. 10-Left, $h^\star$ is inversely correlated to $h_{\text{MLP}}$ and is not correlated (i.e. balanced) to $h_{\text{RN}}$. We know training an MLP with ERM on $D_t$ will choose $h_{\text{MLP}}$. Therefore, D-BAT will perform well (i.e., find $h^\star$) by minimizing its diversification loss. On the contrary, training a ResNet with ERM on $D_t$ will choose $h_{\text{RN}}$. Therefore, as shown in Sec. 4, D-BAT cannot perform well by minimizing its diversification loss. The opposite conclusion holds for Tab. 10-Right.

- **Step 5**: we take $D_t$ and $D_u$ (which are around 12k and 24k images, respectively) and run D-BAT (Pagliardini et al., 2023) (the labels on $D_u$ are inaccessible), and measure the test accuracy on hold-out $D_{\text{ood}} \sim D_u$, which is shown in Tab. 2-Left.

The construction process is thus a *white-box* process (or attack), similar to adversarial attacks, but on the architectural inductive bias aspect. We reiterate that the purpose of this example is to illustrate that the choice of architectural inductive bias can have a very drastic influence on the behavior of diversification methods and this choice is co-dependent on the properties of the unlabeled OOD data.

Additionally, we can demonstrate this co-dependence in a "fine-grained" manner, as shown in Fig. 15. Here we still keep the training data the same (according to $D_t$ in Tab. 10), and construct the unlabeled OOD data $D_u$ such that the spurious ratios of $h_{\text{MLP}}$ and $h_{\text{RN}}$ gradually switch from low to high or high to low. Hence, the Tab. 10-Left and the Tab. 10-Right correspond to the left and right extremities of the x-axis of Fig. 15, and in between are interpolations between the two distributions (spurious ratios). This gives a better view of how one inductive bias gets favorable through the transitions of spurious ratios, and vice versa.

| $D_u$ | $\mathcal{A}$ | Test Acc.(%) |
|-------|---------------|--------------|
| $h^\star \perp h_{\text{ViT}}$ | ViT | $76.4_{\pm 0.6}$ |
|       | ResNet18 | $58.1_{\pm 0.4}$ |
| $h^\star \perp h_{\text{RN}}$ | ViT | $68.1_{\pm 0.3}$ |
|       | ResNet18 | $79.0_{\pm 0.5}$ |

Table 11: **Tab. 2 with another architecture pair.** ViT far exceeds ResNet18 when $h^\star \perp h_{\text{ViT}}$ and vice versa. Results are averaged over 3 seeds.

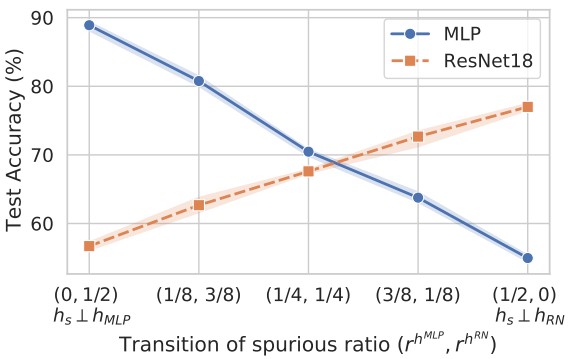

Figure 15: **The favorable inductive bias changes over transitions of spurious ratios**. As the unlabeled OOD data $D_{\text{ood}}^U$ changes, the optimal inductive bias switches from MLP to Resnet18, indicating their co-dependence. Results are averaged over 3 seeds.

