# OpenReview forum: "Unraveling the Key Components of OOD Generalization via Diversification"
_ICLR.cc/2024/Conference — ICLR 2024 poster_

### Official Review · Reviewer_WG9i · 2023-10-31

**Soundness:** 3 good
**Presentation:** 2 fair
**Contribution:** 2 fair
**Rating:** 5
**Confidence:** 3

**Summary:**

This paper studies the class of  “diversification” methods that address OOD problem, and identify the key components contributing to their performance. Some findings are provided through research，that can help guide the development of approaches. Some experiments are conducted to support their results.

**Strengths:**

1) This paper investigates the “diversification” methods from theoretically from different aspects. Some key results are provided.
2) Valid experiments have proved the rationality of their views.
3) Some showcase are provided to validate the meaning of the findings.

**Weaknesses:**

1) What are the unique advantages of “diversification” methods to solve OOD problems？Does it contribute much to OOD community to study the components of “diversification”? More discussion of motivation and related works are needed.
2) Despite the limitation talked in paper, I think experimental setup of the paper is still simple. More complicated diversification loss, approaches, need to be included.

**Questions:**

In addition to the methods used in the paper, does it need to consider more models to support the conclusions of the paper?

**Details Of Ethics Concerns:**

nan

---

> ### Author Response · Authors · 2023-11-16
> **Response to Reviewer WG9i**
>
> We thank the reviewer for their time reviewing and providing useful feedback, we kindly answer raised questions as follows.
>
> > What are the unique advantages of “diversification” methods to solve OOD problems？Does it contribute much to the OOD community to study the components of “diversification”?
>
> We kindly refer to the ‘The focused scope of our work’ section in the general response for a detailed answer. We will be sure to add more discussion in the next revision.
>
> > More complicated diversification loss, approaches, need to be included. In addition to the methods used in the paper, does it need to consider more models to support the conclusions of the paper?
>
> At the time of writing, to the best of our knowledge, there were no other diversification losses or approaches that are fundamentally different from D-BAT and DivDis. The two methods differ not only on diversification losses but also on the optimization strategies, thus representing a spectrum of possible dynamics in this learning framework well.
>
> Throughout the paper, we convey crucial messages that provide insights to guide the future design of diversification methods and the usage of current ones, and each message is supported by theoretical proofs (proposition 1, proposition 2), and experimental verifications as below:
>
> - For proposition 1, we further justify the performance of diversification methods is highly dependent on spurious ratio by firstly verifying the solution of synthetic 2D illustrative example and secondly showing the conclusion holds in practice with datasets in different scales (M/C, M/F, and CeleA-CC).
> - For proposition 2, we show with 10 models (spanning different training strategies) that diversification alone cannot lead to OOD generalization the inductive bias brought by the used learning algorithm is important.
>
> For the reasons above, we believe our conclusions are well-supported. However, we would be very glad to provide further evidence, but was wondering which kind of models or approaches the reviewer would be interested to see?

---

> > ### Author Response · Authors · 2023-11-21
> > **Hoping that our response could address your concern**
> >
> > Dear Reviewer WG9i,
> >
> > Thank you again for your time and effort in reviewing our work! We would appreciate it if you can let us know if our response has addressed your concern and thus improved your assessment of our paper. We look forward to hearing from you and remain at your disposal for any further clarification that you might require.

---

> > ### Comment · Reviewer_WG9i · 2023-11-22
> >
> > Thanks for the clarifications. After rebuttal, I will maintain my initial score.

---

> > > ### Author Response · Authors · 2023-11-22
> > >
> > > Dear reviewer WG9i,
> > >
> > > Thank you for the response. In our rebuttal, we provided an itemized response to each of your questions. Specifically, we provided clarification on 1) the advantages of diversification methods (presented in the general response), 2) why the experimental setup in our paper is not simple, and 3) why we believe our conclusions are well-supported.
> > >
> > > Could you please comment if you did not find these clarifications satisfying and how we can further improve our paper?
> > >
> > > Thank you!

---

### Official Review · Reviewer_MHwE · 2023-10-31

**Soundness:** 3 good
**Presentation:** 3 good
**Contribution:** 2 fair
**Rating:** 6
**Confidence:** 3

**Summary:**

The authors study 2 recently proposed supervised learning "diversification" methods (DivDis and D-Bat) which aim to achieve high accuracy out-of-distribution (OOD) by generating a diversity of hypotheses each of which fit the training data but which disagree on additional unlabelled data from a different distribution  and then picking one final hypothesis from the list . The process is intended to reduce the odds of yielding a model which relies on spurious correlations which would not persist under distributional shift.  Through a mix of theory , toy experiments and real-world-data experiments, the authors arrive at a number of findings which warn that neither DivDis nor D-Bat (nor any particular diversification method ) is likely to work well in all situations. The performance of diversification methods is shown to be highly sensitive to the distribution of the unlabelled data on which diversity is measured.  There is an interaction between the learning algorithm and the unlabelled data distribution such that each affects the optimality of the other. The appropriateness of the inductive bias of the learning algorithm is shown to be critical to the success of diversification methods. Increasing the number of hypotheses generated by the diversification methods does not necessarily fix these concerns...the interplay between learning algorithm and unlabelled distribution and the importance of inductive bias persist.

**Strengths:**

The work is original to the best of my knowledge.

The paper is well written and clear. I have a few suggested typo-style edits in the weaknesses section, but the writing is certainly strong.

I did not notice any errors or incorrect conclusions in the findings. Although I didn't have time to go through every result in extreme detail, I have a reasonable amount of confidence in the correctness of the results, generally.

Generalization which is robust to out-of-distribution shifts is certainly a worthy topic.

**Weaknesses:**

My main concern with the paper is whether its results are significant enough to merit acceptance at ICLR. I'm open to being persuaded that the paper is significant enough, but that's not clear to me, for a few different reasons.  I offer these concerns with only moderate confidence, since I am not an expert specifically on distributional shift literature.

The paper focuses largely on the pitfalls and limitations of 2 papers from ICLR 2023, the Lee DivDis and the Pagliardini D-Bat approaches. I don't doubt that these papers are high-quality and significant, but they simply haven't been around long enough to know for sure how significant it is to critique them and scrutinize their flaws. I'm not saying that critiquing them is unworthy- it's just hard to tell whether it's a highly significant contribution.

Also, although the authors do use real-world data to an extent, I find the applications to still be a bit contrived and not illustrative of a clear real-world situation where distributional shift needs to be handled and cannot be avoided.  The MNIST-CIFAR concatenations and Waterbirds-CC are both contrived, i.e., constructed to have spurious correlations, rather than spurious correlations arising naturally.  Office-Home with Art, Product, Clipart, Real-World is somewhat better, but nonetheless, the authors of that dataset did manage to assemble all four sources and so the sensible thing to do would seem to be to train on a dataset with all 4 sources mixed together. I can see how you could have a real-life situation where e.g. only Art, Product, Clipart are available and then you have to design a system which peforms well on Real-World, but the paper would be stronger if a problem domain was studied where that was a real constraint that made distributional shift unavoidable. The most obvious scenario would be distribution shift over time, e.g. , we had to collect images during sunny weather in the summer and then had to deploy the system in darker weather when it was raining...something along those lines.

My other significance concern is the findings (while common-sense) seem a bit unsurprising to me. We've  known since the 1990s from the Wolpert No Free Lunch theorems, the bias-variance tradeoff, etc  that the right inductive bias is crucial to supervised learning success, in general. While it's worthwhile to illustrate a version of this principle for these recently-developed diversification methods, the finding that the principle holds seems more like common sense to me than like a surprising finding that advances the field in a major way. Maybe I'm wrong, though, and maybe I"m expecting too much from a paper worthy of ICLR acceptance.

Another significance concern I have comes from a reaction I had to a sentence on page 4: "We, therefore, focus on studying the first stage and assume access to the oracle that chooses the best available hypothesis in the second stage". While I can see that it's true that the first stage is crucial to the success of the whole diversification approach, as someone who is not that familiar with diversification methods or OOD-robust algorithms, it's not that clear to me that we can assume that the second stage will succeed if the first stage succeeds. I guess the idea is that if K is small, then we only need a modest amount of supervised holdout data to choose from among the K diverse hypotheses generated, e.g. the in-sample bias of picking from among a low-K list using validation data is modest? I suppose that could be true, but I would appreciate more background on that topic.

Given what I wrote in the previous paragraphs and given the focus on 2 papers just published in 2023, the whole paper leans a little bit too far in the direction of assuming that what the authors are studying is very important rather than making the case for the importance of the topic to someone who isn't already deeply involved in OOD and diversification.

Typos:

page 5 function. And the training distribution -> function, and the training distribution (don't start a new sentence here)
page 9 "This is in consistence with" -> This is consistent with

**Questions:**

Can you add some more background for how OOD concerns can arise in industry? Even though the experiments use real world data in some sense, the tweaking of the real world data to introduce spurious correlations makes the experiments a bit less compelling in terms of real-world evidence.

Can you explain a bit about why I can assume that stage 2 (disambiguation ) probably is doable? I need to be reasonably confident that stage 2 is also doable in order to care about the success of stage 1.

---

> ### Author Response · Authors · 2023-11-16
> **Response to Reviewer MHwE (1/2)**
>
> We thank the reviewer for their time reviewing and providing useful feedback, we kindly answer raised questions as follows.
>
> > The focused methods of our work
>
> We would like to clarify that our analysis shows unique axes that make diversification methods work, and is actually acknowledging the significance (rather than criticizing) and providing further investigation and evidence to support these methods to stand the test of time. We kindly refer to the ‘The focused scope of our work’ section in the general response for a detailed answer.
>
> > The significance of the no-free-lunch results
>
> We kindly refer to the ‘The significance of the no-free-lunch results’ section in the general response to all reviewers.
>
> > Concerns on the datasets and problem setup we considered / Can you add some more background for how OOD concerns can arise in the industry?
>
> We kindly answer the two questions together. As shown in [1, 2], when multiple predictive features are present in the training set, neural networks tend to learn the simple (but can be spurious) feature rather than the true causal features because of their ''simplicity'' inductive bias. The case where multiple predictive features co-exist but only the true feature is generalizable is called spurious correlation. This can cause real-world and industrial failure cases, for example:
>
> - In a chest X-ray dataset [3], many images of patients with pneumothorax include a thin drain used for treating the disease. An empirical risk minimization (ERM) model trained on such a dataset can erroneously identify such drains as a predictive feature of the disease, while it's actually a spurious feature. A good deployable model for the model would require sensitivity to physiological signals while being invariant to environmental or operational signals that can change between deployment contexts. (please see https://wilds.stanford.edu/datasets/ for more real-world examples)
>
> which is especially crucial when the reliability of the model is expected to be high, such as in medical/financial/self-driving fields. This has motivated the community to study spurious correlation and the problem has become a main branch of OOD generalization problem and is long-standing, well-defined, realistic but challenging. Following the many existing methods (e.g., [5, 7, 8]), both D-BAT & DivDis and our work use the *well-established* and *real-world* baselines and benchmarks (CelebA, WaterBirds, Camelyon17, etc). Going one step further, D-BAT and DivDis also try to handle a more difficult case, where the spurious feature/hypothesis *always* correlates with the true feature/hypothesis, and this is important because:
>
> - Sometimes it's hard to acquire minor group samples due to temporal or spatial reasons (e.g., a particular species can only be observed in a specific time/location).
> - Therefore, the model has to work on this harder case in order to reliably work on simpler cases.
> - As an example, the medical images of a specific disease tissue are solely available in a hospital/research lab, and the medical model fails to recognize the disease because it learns to only look at the hospital token rather than the tissue.
>
> For our purpose, using MNIST-CIFAR and Waterbirds-CC unlocks more controlled experiments (e.g., a more fine-grained analysis under various spurious ratios) to simulate the different scenarios that could arise in the industry, which makes it possible to understand the conditions required for the diversification methods to work, which, in turn, allows for developing more reliable and powerful methods for real-world applications.

---

> > ### Author Response · Authors · 2023-11-16
> > **Response to Reviewer MHwE (2/2)**
> >
> > > It's not that clear to me that we can assume that the second stage will succeed if the first stage succeeds
> >
> > There are multiple ways to ensure a successful second stage, as also discussed in DivDis.
> >
> > - One solution is to have a small amount of labeled data to disambiguate the hypotheses from the first stage. The data samples to be labeled can either be chosen at random or be chosen by the highest distance of head prediction logits (a larger difference in predictions indicates more information for disambiguation). Having a small amount of validation data is realistic and also adapted by other existing methods[4, 5] for OOD, in order to tune their hyperparameters.
> > - Another solution could be to have additional human supervision, even on the source data with spurious correlation. In particular, a human practitioner can manually observe the relevant region (or word in terms of text) for each hypothesis (parametrized by neural network) by, e.g., Grad-CAM [6], and select the hypothesis best aligned with human understanding of the task.
> >
> > All of the aforementioned solutions require only a few efforts as the number of hypotheses is supposed to be small. This makes the second stage more controllable than the first stage, which needs to reduce all possible hypotheses to a small number. We, therefore, focus on the first stage that provides a necessary condition for the whole method to work. At the same time, we recognize the importance of further research on the second stage in the future.
> >
> > Finally, we thank the reviewer for the suggestion and for noticing the typos, we will include more discussion to highlight the importance of the topic and make a smoother transition for people who are not already deeply involved in OOD and diversification. We will also correct the typos in the next revision. We hope we have addressed your questions, and we remain at your disposal for any further questions you may have.
> >
> > [1] Geirhoss et al., Shortcut Learning in Deep Neural Networks. Nature Machine Intelligence 2020
> > [2] Scimeca et al., Which Shortcut Cues Will DNNs Choose? A Study from the Parameter-Space Perspective, ICLR 2022
> > [3] Oakden-Rayner et al., Hidden stratification causes clinically meaningful failures in machine learning for medical imaging, In Proceedings of the ACM conference on health, inference, 2020
> > [4] Nam et al., Learning from failure: Training debiased classifier from biased classifier, NeurIPS 2020
> > [5] Liu et al., Just train twice: Improving group robustness without training group information, ICML 2021
> > [6] Selvaraju et al., Grad-cam: Visual explanations from deep networks via gradient-based localization, CVPR 2017
> > [7] Sagawa et al., Distributionally Robust Neural Networks for Group Shifts: On the Importance of Regularization for Worst-Case Generalization, ICLR 2020
> > [8] Zhang et al., Correct-N-Contrast: A Contrastive Approach for Improving Robustness to Spurious Correlations, ICML 2022

---

> > > ### Comment · Reviewer_MHwE · 2023-11-20
> > > **Thanks for the response**
> > >
> > > I have read the rebuttal of the authors. I appreciate the explanation regarding the contrived datasets...that it allows for control over the spurious ratio. I will raise my score to a 6.

---

> > > > ### Author Response · Authors · 2023-11-22
> > > > **Thank you**
> > > >
> > > > Dear reviewer MHwE,
> > > > Thank you for the reply. We are glad that our response has addressed your concerns.
> > > > The new score doesn’t seem to be updated in the system (as shown on our side), could you please double-check it?
> > > > Thank you!

---

> > > > > ### Comment · Reviewer_MHwE · 2023-11-22
> > > > > **Fixed it- changed to 6**
> > > > >
> > > > > Changed my score to 6, apologies for not doing that the first time.

---

> > > > > > ### Comment · Reviewer_MHwE · 2023-11-22
> > > > > > **Not a first time reviewer**
> > > > > >
> > > > > > FYI, I checked the 'first time reviewer' box by accident. I am not a first time reviewer, but it won't let me uncheck it.
> > > > > >
> > > > > > Maybe the UI design should be changed there. It's an easy mistake to make, since the default behavior for "code of conduct" next to it is to check that box.

---

### Official Review · Reviewer_xY3D · 2023-11-02

**Soundness:** 4 excellent
**Presentation:** 4 excellent
**Contribution:** 3 good
**Rating:** 6
**Confidence:** 4

**Summary:**

The paper critiques recent trend of building diverse models and then selecting one at test time, as a way of enhancing OOD generalization. It presents empirical and theoretical results on why the methods may not always work.

**Strengths:**

I really like this paper. Given so many methods being proposed for OOD generalization, it is important to take a step back and analyze which ones are likely to work and under what conditions.

This paper finds that the literature on diversification of hypotheses may not be conceptually well-motivated. The key result is that the success of this technique depends on inductive bias of the model architecture and the same architecture may not work well for different kinds of test set.

In hindsight, many of the observations seem obvious. for example, given that the methods use 2-3 different hypotheses, they obviously are relying on the training procedure's inductive bias (otherwise how can 2-3 samples explore the full space of "good" hypotheses?). But still, the authors do a good job of articulating multiple such issues in a single paper.

**Weaknesses:**

While the analysis is compelling, I'm wondering whether these limitations matter in practice. What if we do model selection over multiple architectures and multiple diversity algorithms? Is the risk that the results we get on a cross-validation set may not generalize to the test set?

If so, what are the summary statistics of the test set that the above procedure would need to know? For example, if the spurious ratio of the (unseen) test set is known, can that be used to simulate a pseudo-test set and then do model selection over it?

Overall, I'm unclear of main takeaway of this paper. Should we not use diversification algorithms? What is the better alternative?
Personally, I feel a better message from the paper can be that it identifies the axes on which model selection should be done, if some summary statistics of the test set can be provided in advance.

**Questions:**

See the weaknesses above.
In particular, can the authors setup a model selection expt where the spurious ratio of the unseen test set is known and the test set is simulated. Would it always lead to the correct model? What else do we need to know about the test set? Can that be summarized, or is that not possible (and hence there is no way to know the best model apriori)?

---

> ### Author Response · Authors · 2023-11-16
> **Response to Reviewer xY3D (1/2)**
>
> We thank the reviewer for their time reviewing and providing useful feedback, we kindly answer raised questions as follows.
>
> > Given that the methods use 2-3 different hypotheses, they obviously are relying on the training procedure's inductive bias (otherwise how can 2-3 samples explore the full space of "good" hypotheses?)
>
> Indeed, it is expected that one needs to outline many hypotheses to considerably explore the hypothesis space (as shown in proposition 2). However, D-BAT and DivDis were shown to work in practice with only two hypotheses, making us curious about the components (we study inductive bias and data distribution property) that they rely on to confine the hypothesis space enough to achieve this efficiency. We investigate this and show that the spurious ratio, architecture, and pretraining strategy are important components of the success of these methods. Furthermore, we show the influence of each component (Sec 4 and Sec 5) and their interaction (Sec 5.3).
>
> > What if we do model selection over multiple architectures and multiple diversity algorithms? Is there risk that the results we get on a cross-validation set may not generalize to the test set?
>
> As shown in Figure 3, diversification methods do not work well without pretraining, so the model selection boils down to choosing a diversification loss and a pre-trained model (which includes pretraining strategy and architecture).
>
> As mentioned above (also shown in Figure 3), the pretraining performance on ImageNet is surprisingly not a good proxy for selecting the pretraining strategies. However, a validation set can be used as one of the ways of model selection, and the risk of results' generalization to the test set depends:
>
> - When the validation set is i.i.d to the test set, we can safely do the model selection. Through this work, we have provided important axes (diversification loss, pretraining methods, and their combination) to identify the best model.
> - In terms of cross-validation, i.e., assuming the validation set is i.i.d to the training set and not to the test set, it is not possible to do the model selection. To the best of our knowledge, existing methods in the field do not consider this case and there is generally no solution for this setting.
> - Finally, when the validation set is i.i.d to the unlabeled data, which may be non-i.i.d to the test set, it depends on how many spurious features are present in the dataset.
>     - When there is only 1 spurious feature, knowing the spurious ratio of the test set can help in simulating the corresponding validation set, and the model selection can be done across diversification loss, architectures, and pretraining strategies.
>     - When there is more than 1 spurious feature, knowing such summary statistics is still good, but there is a risk that the results may not generalize to the test set. (see further discussion in the next answer)
>
> > What are the summary statistics of the test set that the above procedure would need to know? For example, if the spurious ratio of the (unseen) test set is known, can that be used to simulate a pseudo-test set and then do model selection over it?
> >
>
> As discussed above, the number of spurious hypotheses and the spurious ratio of each of those hypotheses are good summary statistics to know. In most cases, the number of spurious hypotheses is equal to 1 and there will therefore be only one spurious ratio.
>
> However, as shown in Sec 5.3, when there are more than 1 (2 in this case) spurious hypotheses ($h_1$ & $h_2$) present in the training set, things can be a bit different. Having the spurious ratio for each spurious hypothesis may not help, as different types of the model may latch on a different spurious hypotheses because of their inductive bias, and it’s known to be hard to measure such inductive bias apriori. Thus, the given spurious ratio may not be valid to simulate a pseudo test-set which is valid for all models.
>
> Although there is risk in some cases where no good model selection strategy is available, we would like to highlight and clarify that, the efforts done in this work are focused on the diversification stage (first stage of the two-stage framework shown in Sec 3.2). In cases where there is no reliable way of model selection, which can be known when given the above summary statistics, an ultimate way of model selection can be considering all the axes (diversification loss, pretraining, architecture) above and conducting the second stage (the disambiguation stage) with the hypotheses produced by spanning those axes. Therefore, in the second stage, the best diversification loss & pretraining strategy can still be easily selected with a small amount of additional human supervision, e.g., labeled data points or human observation on the Grad-CAM [1] feature map. This means the diversification method is still useful but may have to pass more burden to the second stage in order to reliably find the hypothesis that is close to the true one.

---

> > ### Author Response · Authors · 2023-11-16
> > **Response to Reviewer xY3D (2/2)**
> >
> > > Should we not use diversification algorithms? What is the better alternative? I feel a better message from the paper can be that it identifies the axes on which model selection should be done, if some summary statistics of the test set can be provided in advance.
> >
> > It is sensible to have multiple diverse hypotheses to tackle the spurious correlation problem, as the model does not know apriori which feature is semantic and which feature is spurious. Thus, we think that diversification methods can be useful in practice. However, further research would be needed to understand them better and develop better versions (e.g., ones that are more robust to the change in the spurious ratio)
> >
> > The purpose of this work is exactly to provide a significant analysis of which axes we need to consider, to what extent (in terms of which statistic and models) we need to consider, and, as discussed above, the potential more efforts required for the second stage (disambiguation) to find the best model (i.e., the model that learns the true hypothesis). This also opens up new directions for developing better diversification methods that take all these axes into account.
> >
> > We hope we have addressed your questions, and we remain at your disposal for any further questions you may have.
> >
> > [1] Selvaraju et al., Grad-cam: Visual explanations from deep networks via gradient-based localization, CVPR 2017

---

> > > ### Author Response · Authors · 2023-11-21
> > > **Hoping that our response could address your concern**
> > >
> > > Dear Reviewer xY3D,
> > >
> > > Thank you again for your time and effort in reviewing our work! We would appreciate it if you can let us know if our response has addressed your concern and thus improved your assessment of our paper. We look forward to hearing from you and remain at your disposal for any further clarification that you might require.

---

### Official Review · Reviewer_shsk · 2023-11-03

**Soundness:** 3 good
**Presentation:** 2 fair
**Contribution:** 2 fair
**Rating:** 6
**Confidence:** 4

**Summary:**

This paper examines two recently proposed algorithms for improving out-of-domain generalization through diverse hypothesis selection with respect to both a labeled and unlabeled dataset. The algorithms try to find hypotheses that agree with the labels on the labeled data but disagree with one another on the unlabeled data.

The authors essentially find that there is "no free lunch" for improving out of domain generalization in that the set of diverse hypotheses selected will depend on which unlabeled data was used to find the diverse hypotheses, which underlying model class was used, and which metric is used to quantify the diverseness of a set of hypotheses.

The diverse hypothesis selection algorithms examined in the paper are:
1. diversify and disambiguate aka "DivDis" (Lee et al. 2023)
2. diverisity by disagreement training aka "DBAT" (Pagliardini et al., 2023)

DivDis and DBAT use different metrics to assess the diversity of a set of hypotheses.

The main contributions of the paper are:
1. Proposition #1 which states that DivDis and DBAT will select different second hypotheses w.r.t. an unlabeled dataset depending on the extent to which the first hypothesis selected (i.e., the ERM) agrees with the true hypothesis on the unlabeled data.
2. Proposition #2 which states that the number of diverse hypotheses generated by an algorithm like DivDis or DBAT needs to be super-linear in the number of unlabeled datapoints in order to ensure that at least one has greater than 50% accuracy with respect to the true hypothesis.
3. Experiments on synthetic and real datasets to support propositions #1 and #2, and which also show that the success of a diverse hypothesis generation algorithm jointly depends on the underlying model class and the unlabeled set of data selected.

**Strengths:**

This paper really dives into the intricacies of the diverse hypothesis generation problem and does a wonderful job illustrating how complex the problem truly it is; that is, success simultaneously depends on all variables. In my opinion, this message should be communicated more frequently in conference proceedings.

In particular, proposition #1 is quite illuminating in that it shows how DivDis and DBAT select different diverse hypotheses from one another, and there are different regimes defined in terms of the agreement with the true hypothesis on the unlabeled data where each is superior. And here, it was nice to see how the experiments on the synthetically constructed datasets (e.g., MNIST/CIFAR) supported the theoretical findings.

**Weaknesses:**

This paper has a number of weaknesses. I found the presentation more confusing and dense than it could be:

  1. In particular, there is some terminology and notation that can be improved for greater understanding. The "spurious ratio" index is a poorly named quantity because it's literally the accuracy of the selected hypothesis with respect to the true hypothesis h*. Namely, h* has the maximum spurious ratio value of 1.0, but it's definitely NOT spurious as it's the true hypotheis. Another name, like the "agreement ratio" would be much clearer.

  2. Similarly, the plots on the left side of figure 2 do not seem to agree with the description of the synthetic problem in the first paragraph of section 4.1, which made it very hard to understand what was being communicated (I elaborate on my confusion below).

And though a good message to repeat, the no-free-lunch findings discussed in the paper are known in the supervised learning setting, so they definitely need to hold for this harder setting with labeled + unlabeled data. Could this research direction become even more constructive by making certain statistical assumptions? In the paper, D_u can be any "out of distrubtion" distribution; and the capacity of the learning algorithm is assumed to be infinite in proposition 2, even though in other parts of the paper, inductive biases of different model classes are highlighted (which suggests that there the model class has less than infinite capacity in practice).

The results for DivDis in table 1, section 5.2 for increasing K are not convincing because alpha needs to increase as the square of the number of hypotheses in the set (i.e., O(K^2)). Otherwise the regularization becomes too strong and dwarfs the empirical risk. But the text makes it seem like a fixed alpha was used, which would needlessly harm DivDis' performance as K gets larger.

**Questions:**

Regarding figure 2, h* is defined in the text such that instances with x1 > 0 should be labeled as positive examples. I assume that x1 is the horizontal axis such that points to the right are positive. But the plot and its legend shows points to the left are positive (Class 1). Is my interpretation correct here?

Assuming it is correct, hyperplanes are typically defined by their normal, in which case is should be that h*(x) = h(x; 0), not h*(x) = h(x; pi/4), since theta=0 radians points in the positive horizontal direction to the right, whereas theta=pi/4 radians points up to the top of the page.

---

> ### Author Response · Authors · 2023-11-16
> **Response to Reviewer shsk**
>
> We thank the reviewer for their time reviewing our paper and providing useful feedback, we kindly answer the raised questions as follows.
>
> > Regarding Weakness 1: the spurious ratio
>
> Thanks for pointing out the confusion. Indeed, this can be confusing and we will carefully consider a better name (e.g., agreement ratio as suggested) in the next revision. The new definition can be:
>
> Definition 1 (Agreement Ratio) Given a hypothesis h, the agreement ratio $r_D^h$, with respect to a distribution $D$ and its true labeling function $h^∗$ is defined as the proportion of data points where $h^∗$ and $h$ agree, i.e., have the same prediction: $r_D^h = \mathbb{E}_{x \sim D}[h^*(x) = h(x)]$.
>
> We also would like to comment that, though $h^*$ has $r_D^{h^*}=1$, this relation is not symmetric, i.e., a hypothesis $h$ with $r_D^{h}=1$ does not mean $h=h^*$. This is because the quantity is defined on a distribution $D$. For example, on $D_t$ where complete spurious correlation exists, there exists a $h$ such that $r_{D_t}^{h}=1$, but on unlabeled data $D_u$, the same $h$ may have $r_{D_u}^{h} \in [0,1]$. In the scope of our work, we mostly examine this quantity on $D_u$, and we assume the test set $D_{ood}$ is a distribution where no spurious correlation exists, i.e., $r_{D_ood}^{h} \approx 0.5$ (so that the test set can reflect how well the diversification methods find the true hypothesis $h^*$). We hope this could address your confusion.
>
> > Regarding Weakness 2: Figure 2 and definition of hyperplanes
>
> As mentioned in the description of Sec 4.1, $x_1$ is the horizontal axis and in this case, the reviewer is correct that $h^*(x)=\mathcal{I}\{x_1 > 0\}$ would be 1 for the points on the right. This is indeed a typo, and we thank the reviewer for noticing it. We will correct it. However it should be noted that this doesn't make any difference in the conclusions made in Proposition 1, and in the behavior illustrated in Figure 2, given that we're in the symmetric binary classification case and the classes are balanced.
>
> For defining the hyperplanes, we wanted to match what D-BAT does (in Figure 17) and therefore used the radian of the classification plane w.r.t horizontal axis $x_1$ as mentioned in Sec 4.1, instead of the normal. We hope this allows for a better readability.
>
> > Regarding the no-free-lunch findings
>
> We kindly refer to the ‘The significance of the no-free-lunch results’ section in the general response for a detailed answer.
>
> > The results for DivDis in Table 1, section 5.2 for increasing K are not convincing because alpha needs to increase as the square of the number of hypotheses in the set (i.e., O(K^2)).
>
> We assume the reviewer means alpha needs to *decrease* (rather than increase) when increasing K in order for the regularization not to become too strong. The reviewer's concerns are correct given Equation 2, as the diversification losses are summed, and the value is not normalized w.r.t the number of hypotheses K. In practice, however, we always *average* the diversification losses instead of summing them up, which results in having $\alpha_K = \frac{\alpha}{K(K-1)}$. The $\alpha$ in the numerator is what we keep fixed in practice, which corresponds to scaling the loss by a factor of $~1/K^2$ as the reviewer rightfully suggested. We thank the reviewer for noticing this confusion. We will update the loss function to reflect the normalization term.
>
> We hope we have addressed your questions, and we of course remain at your disposal for any further questions you may have.

---

> > ### Author Response · Authors · 2023-11-21
> > **Hoping that our response could address your concern**
> >
> > Dear Reviewer shsk,
> >
> > Thank you again for your time and effort in reviewing our work! We would appreciate it if you can let us know if our response has addressed your concern and thus improved your assessment of our paper. We look forward to hearing from you and remain at your disposal for any further clarification that you might require.

---

> ### Comment · Reviewer_shsk · 2023-11-23
> **Replying to authors' response to my review**
>
> Thank you for replying to my questions and concerns. Here are my responses:
>
> 1. Regarding the spurious ratio, I had understood that r_D^(h) depends on D and that r_D^(h) can be 1 even when h != h*.  But thank you for verifying I was aware of this.
>
> 2. On figure 2 and the definition of hyperplanes, thank you for confirming my assumption and dispelling my confusion. I had stared at that figure for a long time :-)
>
> 3. Regarding alpha and table 1, that you for clarifying that there was a hidden correction for the number of hypotheses, "K". I am now convinced for the results in table 1.
>
> 4. Finally, on the NFL, you state that "your results do do not directly follow from the original NFL, similar to the results that researchers in the meta-learning field [2,3] or other fields [4,5] obtained, which can also be seen as expected in hindsight."
>
> But actually I think they do follow from NFL. Imagine that a diversification method existed called "DivBest" was in fact sufficient for OOD generalization.  Then one could have a free lunch in the supervised setting by:
> i) Collecting a large training set of labeled examples.
> ii) Discarding half or more of the labels to form an unlabeled data set.
> iii) Running DivBest on the labeled + unlabeled data to learn a model \hat{M} with DivBest's OOD generalization guarantees.
>
> You have addressed 3/4 of my concerns so I will revise my overall score upward.
>
> The larger concern that "results follow from the no-free-lunch theorem" still seems open to me. Or, if the results do not follow from NFL, the paper in its current form does not make it clear why. If the authors have insights here, I'd be appreciate if they could share them.

---

> > ### Author Response · Authors · 2023-11-23
> > **Additional clarifications on the difference with NFL**
> >
> > Dear reviewer shsk,
> >
> > We thank you for acknowledging our clarifications and are glad they addressed most of the concerns. We also thank you for providing additional details on one of your questions about NFL, which we further address below.
> >
> > If we consider a two-stage diversification method as one learning algorithm that outputs only a single hypothesis, then we agree that NFL is sufficient. However, this explanation does not consider the specific structure of diversification methods, which is 1) two-staged, where 2) the 1st diversification stage outputs multiple hypotheses K. **In contrast, our result takes this structure into account and does not follow directly from NFL. Below, we share our insights.**
> >
> > 1. As discussed in our rebuttal and the DivDis work, the 2nd stage does not need to be limited to statistical learning, and, e.g., human feedback (e.g., using Grad-CAM or other explainability methods) can be used to choose the resulting hypothesis out of K. Moreover, the 2nd stage can be applied to different downstream applications, e.g., in the Waterbirds case, one can choose between background or bird class predictors by only changing the 2nd stage. **Therefore, it is important to understand the performance of the 1st stage alone, which needs to output K “most useful” hypotheses (instead of a single one in the standard supervised learning setting). NFL does not cover this scenario**
> > 2. Our Proposition 2 focuses precisely on the 1st diversification stage and **provides an understanding of how its performance scales with K**, even when considering the best-case scenario for the 2nd stage, and **suggests that the 1st stage is the most crucial one** for the success of a diversification method as a whole, and it is more beneficial to focus on improving this stage instead of the 2nd one.
> >
> > Overall, this theoretical result motivates our empirical study, where we find that in practice (Tab. 1), scaling K does not improve the performance of the whole diversification algorithm even when considering the 2nd stage to be an oracle, and the choice of the inductive bias, e.g., in the form of the architecture and pretraining method still plays a crucial role (Fig. 3 and Tab. 1).
> >
> > We thank you for your comments, and we will incorporate these points into our updated manuscript to make it more clear.

---

### Author Response · Authors · 2023-11-16
**To All the Reviewers (1/2)**

We thank the reviewers for their time, thoughtful reviews, and suggestions that will help us further improve the manuscript. We appreciate the reviewers found it important to perform this type of analysis in our work and found the message should be communicated more frequently in conference proceedings. We also thank the reviewers for acknowledging our experiments as compelling, our theoretical/experimental results as illuminating, and our writing as clear. We have provided detailed responses for each reviewer, and we provide a summary here.

Through our work, we provide a critical analysis on two recent works (we refer to diversification methods, see ‘the focused scope’ section below for more details). Our investigation shows that the success of these diversification methods crucially relies on the sweet spot of spurious ratio,  architecture/algorithm inductive bias (see ‘the significance of the no-free-lunch results’ section below for more details), and their interactions. This reveals crucial axes and summary statistics that need to be taken into consideration before applying existing diversification methods or designing new ones for OOD generalization and potentially opens up new research directions, e.g., spurious ratio invariant diversification methods.

## The focused scope of our work

As high-score papers (8888 and oral for D-BAT, 8886 for DivDis) in ICLR'23, D-BAT, and DivDis focus on the spurious correlation or underspecification problem, which is an important branch of OOD generalization problem and a long-standing, well-defined, realistic but challenging problem. Based on this, they provide unique advantages in several aspects:

- They are shown to be the SOTAs across classic benchmarks (i.e., the datasets we considered in our work) when compared to existing methods for spurious correlation and unsupervised domain adaptation methods.
- They are more deployable since they don't need additional annotation, that is, group labels, to tune hyperparameters.
- They can handle a more difficult setting, compared to previous methods, where the spurious feature completely correlates with the true feature.
- They jointly open up a new direction that captures a diverse set of predictive features rather than a single one, for spurious correlation problems.

It is then a natural follow-up to investigate the reasons for these diversification methods to show SOTA results under an even more challenging setting (where the complete spurious correlation may exist) while requiring less information (only unlabeled data points) than previous methods (e.g, group annotation, as discussed in related work). Indeed, the fact that they require less information results in the methods relying more on a “good” combination of unlabeled data and model,  and we characterize precisely what constitutes “good” in this context. Thus, our results reveal the key components/axes for these methods to work and the caveats that we should keep in mind before applying them blindly. We believe (also acknowledged by reviewer shsk, xY3D) that this kind of analysis is critical for the OOD community to understand current methods and develop much better methods.

---

> ### Author Response · Authors · 2023-11-16
> **To All the Reviewers (2/2)**
>
> ## The significance of the no-free-lunch results
>
> We would like to clarify the significance of our results on the insufficiency of diversification alone for OOD generalization, i.e., the no-free-lunch result.
>
> 1. Indeed, this result might be expected given the no-free-lunch theorem for the standard supervised setting [1]. However, it does not directly follow from the original NFL, similar to the results that researchers in the meta-learning field [2,3] or other fields [4,5] obtained, which can also be seen as expected in hindsight.
> 2. We would like to note that we present this result not as a mere critique of the diversification methods but as a meaningful step in studying them from the theoretical perspective. It suggests that there is no significant benefit in diversification-only from the *theoretical* perspective.
> 3. However, despite neural networks being universal approximators [6], and NFL suggesting that each task requires a specific inductive bias, neural networks are successful in practice.  Similarly, our theoretical result does not need to translate into poor generalization in practice. We, therefore, follow by a more holistic experimental analysis of these diversification methods and study how they perform in practice. We show that while being successful in some scenarios demonstrated in the original papers, the diversification mechanism does not bring benefits on top of standard ERM training in more pathological situations (e.g., as shown in Tab. 1, and Sec. 4).
>
> Based on the no-free-lunch suggested in our theoretical results, our findings provide overlooked factors to be aware of in order for diversification methods to achieve good results, and we clarify that in practice the actionable point for applying existing diversification methods is to jointly consider the axes we identified (data spurious ratio, model choices, etc) and refer to the disambiguation stage (as mentioned in Sec 3.2) for the best model.
>
> [1] Wolpert et al., No Free Lunch Theorems for Optimization, IEEE Transactions on Evolutionary Computation, 1997
> [2] J. Baxter, A Model of Inductive Bias Learning, Journal of artificial intelligence research, 2000
> [3] J. Baxter, Theoretical Models of Learning to Learn, in Learning to Learn, 1998
> [4] Bendidi et al., No Free Lunch in Self Supervised Representation Learning, 2023
> [5] Hanneke et al., A No-Free-Lunch Theorem for Multi-Task Learning, The Annals of Statistics, 2022
> [6] Hornik et al., Multilayer Feedforward Networks are Universal Approximators, Neural Networks, 1989

---

### Meta-Review · Area_Chair_QZ3m · 2023-12-12

**Metareview:**

This paper discusses existing diversification methods for handling OOD generalisation.  It attempts to characterise when such methods are best suited to work. It thus brings together a new perspective which would greatly help in deploying such methods for spurious correlation.

**Justification For Why Not Higher Score:**

The paper should have explored potential new avenues on when such methods fail.

**Justification For Why Not Lower Score:**

The paper contributes substantial insights on when the existing Diversification methods and when are they likely to fail. This would be of  interest to ICLR community.

---

### Decision · Program_Chairs · 2024-01-16

Accept (poster)